# Genome-wide association study of pulpal and apical diseases

Aino Salminen [1] ✉, Kati Hyvärinen[2], Jarmo Ritari [2], Jussi M. Leppilahti[3,4,5], Ulla Palotie [1], Ville Vuollo[3,5], Oleg Kambur[6,7,8,9], FinnGen*, Estonian Biobank Research Team*, Kadri Reis[10], Anu Reigo [10], Priit Palta [10], Markus Perola[6], Juha Sinisalo [11], Aki S. Havulinna [6,12,13], Päivi Mäntylä[7], Ulvi Kahraman Gürsoy[14], A. Liisa Suominen [7,15], David P. Rice [1], Vuokko Anttonen[16], Pekka Nieminen [1] & Pirkko J. Pussinen [1,7]

Infections of the dental pulp are common sequelae of microbial activity and host susceptibility, affecting >80% of adult population. We performed a genome-wide association study on endodontic infections utilizing Finnish health registry and genotype data from FinnGen. Cases [132,124 (27.2%)] had at least one ICD10-diagnosis code of pulpal or apical diseases, whereas 353,106 individuals without diagnoses served as controls. We investigated two clinical sub-phenotypes, Pulpitis and Necrosis of pulp or apical periodontitis. Our analysis resulted in significant associations in 12 chromosomes and 15 independent loci, such as those near HORMAD2 gene and those in the HLA region. The imputed HLA alleles, especially DRB1*04:01 and DQB1*03:01, were associated with endodontic infections. Bioinformatic analysis of the top variants indicated several potential regulatory variants which are involved in MHC class II protein complex, humoral immune responses, and antigen processing. Our study widens understanding on how immune dysregulation resulting from immunogenetic variation is involved in the pathogenesis of endodontic infections.

According to the Global Burden of Disease, untreated dental caries is the most common global health concern affecting ~2 billion people (https://www.who.int/health-topics/oral-health). Untreated caries may allow the microbes to enter the root canal system including the pulp chamber of the tooth, which eventually results in pulpal and apical diseases. In such scenario, the principal diseases are pulpitis, the inflammation of the dental pulp, pulp necrosis, and apical periodontitis, inflammation and destruction of the periradicular tissues. The symptoms of these infections can comprise pain, sensitivity to temperature changes, painful response to biting, and tenderness with

[1]Oral and Maxillofacial Diseases, University of Helsinki and Helsinki University Hospital, Helsinki, Finland. [2]Finnish Red Cross Blood Service, Research and Development, Helsinki, Finland. [3]Research Unit of Oral Health Sciences, Faculty of Medicine, University of Oulu, Oulu, Finland. [4]Department of Oral and Maxillofacial Surgery, Oulu University Hospital, Oulu, Finland. [5]Medical Research Center Oulu, Oulu University Hospital and University of Oulu, Oulu, Finland. [6]Finnish Institute for Health and Welfare, Helsinki, Finland. [7]School of Medicine, Institute of Dentistry, University of Eastern Finland, Kuopio, Finland. [8]Department of Internal Medicine, University of Turku, Turku, Finland. [9]Department of Pharmacology, Faculty of Medicine, University of Helsinki, Helsinki, Finland. [10]Estonian Genome Centre, Institute of Genomics, University of Tartu, Tartu, Estonia. [11]Heart and Lung Center, Helsinki University Hospital and University of Helsinki, Helsinki, Finland. [12]Institute for Molecular Medicine Finland (FIMM), HiLIFE, University of Helsinki, Helsinki, Finland. [13]Department of Computing, Faculty of Technology, University of Turku, Turku, Finland. [14]Department of Periodontology, Institute of Dentistry, University of Turku, Turku, Finland. [15]Oral and Maxillofacial Teaching Unit, Kuopio University Hospital, Kuopio, Finland. [16]Research Unit of Population Health, University of Oulu, Oulu, Finland. *Lists of authors and their affiliations appear at the end of the paper. ✉e-mail: aino.m.salminen@helsinki.fi

**Table 1 | Characteristics of the populations**

| Cohort | Phenotype | Cases | | | Controls | | |
|---|---|---|---|---|---|---|---|
| | | Age, mean (years) | Females, n (%) | Males, n (%) | Age, mean (years) | Females, n (%) | Males, n (%) |
| FinnGen-discovery | Pulpal and apical diseases | 60.9 | 76832 (58.2) | 55292 (41.8) | 60.9 | 198271 (56.2) | 154835 (43.8) |
| | Pulpitis | 54.3 | 29820 (62.0) | 18300 (38.0) | | | |
| | Necrosis of pulp or apical periodontitis | 62.3 | 58134 (56.0) | 45698 (44.0) | | | |
| FinnGen - replication | Pulpal and apical diseases | 55.3 | 1890 (45.3) | 2282 (54.7) | 57.4 | 4805 (45.6) | 5728 (54.4) |
| | Pulpitis | 50.2 | 822 (46.8) | 935 (53.2) | | | |
| | Necrosis of pulp or apical periodontitis | 56.7 | 1407 (43.3) | 1840 (56.7) | | | |
| EstBB | Pulpal and apical diseases | 40.2 | 39223 (67.4) | 18939 (32.6) | 45.8 | 99010 (64.6) | 54148 (35.4) |
| | Pulpitis | 29.6 | 11880 (65.2) | 6338 (34.8) | | | |
| | Necrosis of pulp or apical periodontitis | 41.4 | 31486 (68.1) | 14780 (31.9) | | | |
| NFBC 1986 | Deep caries[a] and regular dental pain[b] | 34.2 | 158 (60.3) | 104 (39.7) | 34.1 | 379 (61.8) | 234 (38.2) |
| NFBC 1966 | Deep caries[a] and regular dental pain[b] | 46.4 | 253 (53.6) | 219 (46.4) | 46.4 | 538 (55.8) | 427 (44.2) |
| | Apical periodontitis[c] | 46.4 | 165 (48.4) | 176 (51.6) | 46.3 | 637 (57.1) | 479 (42.9) |

[a]International Caries Detection and Assessment System (ICDAS) 5 and 6. [b] Dental pain fairly or very often. [c] Periapical rarefactions in panoramic radiographs

palpation. However, chronic lesions are often asymptomatic and often detected only in dental radiographs[1]. Symptomatic irreversible pulpitis and apical periodontitis are common reasons for dental emergency visits[2]. It has been estimated that half of the world's adult population have at least one tooth with apical periodontitis[3] highlighting the magnitude of the dental disease burden.

Pulpal and apical diseases not only cause pain, discomfort, and high economic burden for the patient, but they are also among the main causes of tooth extraction resulting in partial or total edentulism. In addition to caries and endodontic infections, common causes for tooth loss are periodontal disease and traumas. Tooth loss is negatively associated with self-image, self-esteem, and quality of life. Indeed, alterations in the functionality of our dentition influence the choice of food that we eat. Low number of teeth is a risk factor for both incident cardiovascular diseases and all-cause mortality[4]. Adding number of missing teeth to established risk profiles improves the risk prediction of incident cardiovascular disease events and diabetes as well as all-cause mortality[5]. Thus, clarifying the mechanisms behind endodontic infections is important for prevention, risk profiling, and discovery of novel treatment targets to further increase wellbeing and health.

Besides hard tissue composition, genetic factors may affect the oral microbial community structure, host response, or the interactions between microbes and host[6]. Genetic variations of matrix metalloproteinase (MMP)−1, −2 and −3[7,8], interleukin 1β[9], and heat shock protein (HSP)−1L and A6[10] have been associated with different endodontic phenotypes using candidate gene approach. Recently, a genome-wide association study (GWAS) of apical periodontitis among patients with deep caries did not identify significant loci[11]. Furthermore, a polygenic risk score from the summary statistics was not associated with endodontic infections in an independent biobank linked with ICD diagnosis codes.

To increase the knowledge on host genetic variants predisposing to endodontic infections, we performed a GWAS on FinnGen participants with diagnosis codes of pulpal and apical diseases derived from Finnish national health registers.

## Results

Phenotype and covariate data were available for 485,230 individuals in FinnGen[12] for the discovery analyses. Number of cases with any of the ICD-10 diagnosis codes for Diseases of pulp and periapical tissues (K04 category, phenotype Pulpal or apical diseases) was 132,124, while 353,106 participants without the diagnoses served as controls in the GWAS. Of the cases, 76,832 (58%) were female and 55,292 (42%) were male (Table 1). The most frequent ICD-10 code was K04.5 (IDC-10 diagnosis Chronic apical periodontitis, $n = 78,933$) followed by K04.1 (Necrosis of pulp, $n = 29,544$), K04.0 (Pulpitis, $n = 27,449$), K04.4 (Acute apical periodontitis of pulpal origin, $n = 23,050$), and K04.08 (Other specific pulpitis, $n = 18,137$) (Supplementary Fig 1). The numbers of cases were 48,120 for phenotype Pulpitis and 103,832 for phenotype Necrosis of pulp or apical periodontitis.

### Discovery GWAS reveals associations in chromosome 22, in the HLA region, and 13 other loci

GWASs of the three phenotypes identified 22 loci with genome-wide significant associations (Fig. 1; Tables 2 and 3): seven for Pulpal and apical diseases, eight for Pulpitis, and seven for Necrosis of pulp and apical periodontitis. Seven of the loci were shared between phenotypes by location and the associated variants. Thus, there were 15 independent loci and 19 lead SNPs to be considered. The Manhattan plots of the GWASs of three phenotypes are presented in Fig. 1 and summary statistics of all variants with genome-wide significance are presented in Supplementary Data 1. The lead SNPs in each genome-wide significant locus are listed in Tables 2 and 3, whereas the locus zoom plots are shown in Supplementary Figs. 2–4.

Strongest evidence for association was identified in a 500-650 kb wide region in chromosome 22 for all three phenotypes (Supplementary Figs. 2–4). Finemapping indicated >60 credible variants for each phenotype and p-values for the two lead SNPs rs9614152 and rs9614155 were below $2*10^{-20}$ (Tables 2 and 3). The alternative alleles were protective against the phenotypes and the population attributable risks (PARs) of the major risk alleles varied from 0.003 to 0.141. The lead SNPs were located in the telomeric end of the genomic region and intronic or downstream of the gene *HORMAD2* (HORMA Domain Containing 2). The nearest genes included *LIF* (leukaemia inhitory factor), *LIF1-AS1* (LIF-antisense-1), *OSM* (oncostatin M), *HORMAD2-AS1*, *MIR6818*, and *MTMR3* (myotubularin related protein 3) with *LIF-AS1* having the nearest transcription start site to the lead SNPs (distances of 72 and 53 kb, respectively).

The second locus shared between the three phenotypes was located within the HLA (human leucocyte antigen) class II region. The

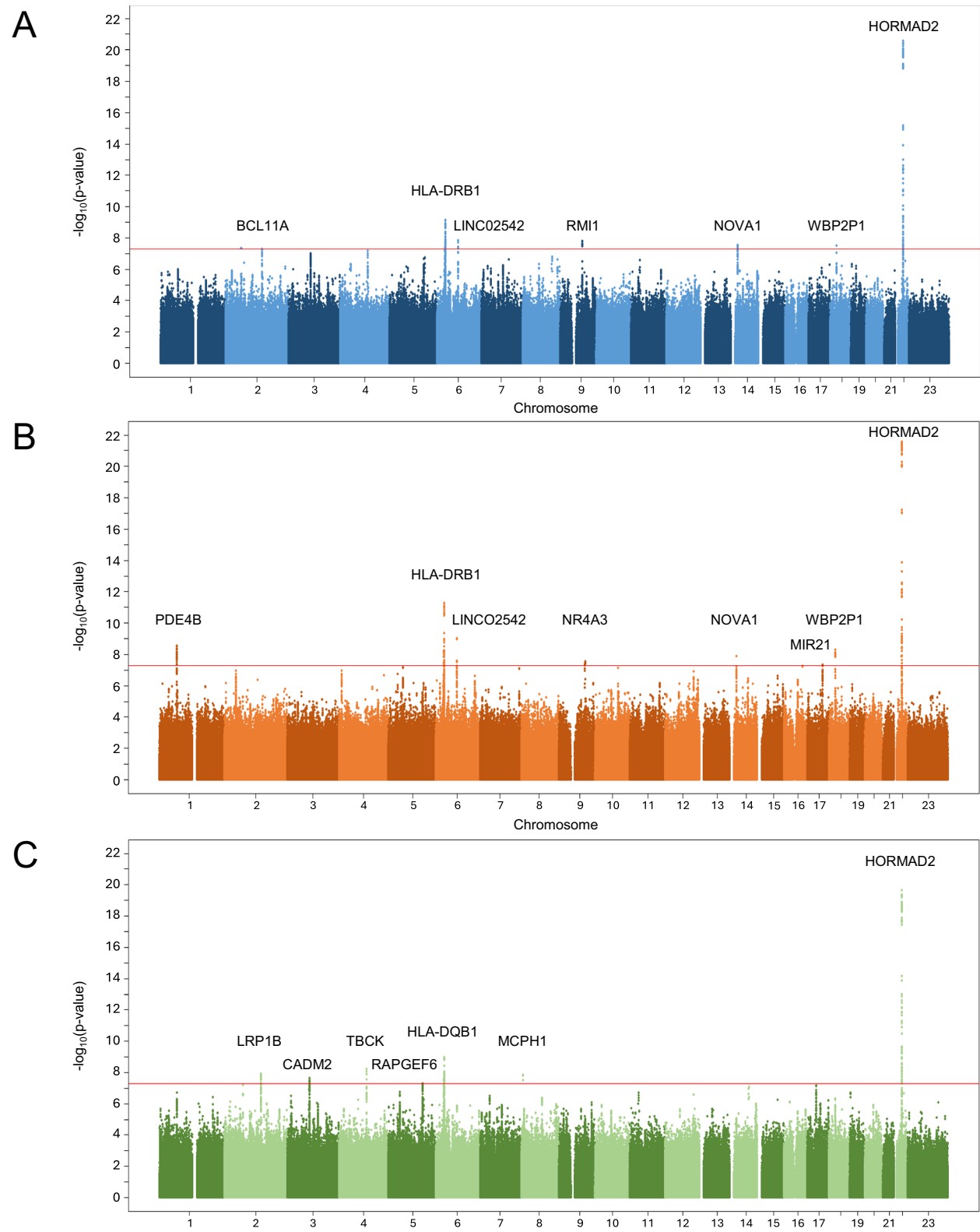

**Fig. 1 | Manhattan plots of the GWAS results in the discovery cohort.** The overall population derived from FinnGen (data release 12) including 485,230 individuals. The phenotypes were constructed based on national health register diagnoses codes. Associations between each SNP and phenotypes were analyzed with REGENIE[44] using an additive model. Each SNP is plotted as a data point with chromosomal position on the x-axis and p-values on the y-axis on a negative logarithmic scale. The genes nearest to the lead SNPs are shown. The red line indicates the threshold for genome-wide significance $p < 5 * 10^{-8}$. **A** Phenotype Pulpal and apical diseases with 132,124 cases and 353,106 controls (genomic inflation factor $\lambda = 1.13$). **B** Phenotype Pulpitis with 48,120 cases ($\lambda = 1.10$). **C** Phenotype Necrosis of pulp or apical periodontitis with 103,832 cases ($\lambda = 1.12$).

**Table 2 | Lead SNPs of the loci associated with endodontic infections in the FinnGen discovery cohort**

| Phenotype | Chr | rsID | Position (GRCh38) | Ref/alt allele | Type | Alt allele frequency | | Beta (SE) | PAR | p-value | Nearest genes |
|---|---|---|---|---|---|---|---|---|---|---|---|
| | | | | | | Cases | Controls | | | | |
| Pulpal and apical diseases | 2* | rs143351662 | 60464983 | G/A | intron | 2.30% | 2.09% | 0.087 (0.016) | 0.003 | 4.48*10^-8 | BCL11A |
| | 6 | rs9270911 | 32604425 | C/T | intergenic | 43.0% | 42.3% | 0.029 (0.005) | 0.016 | 7.70*10^-10 | HLA-DRB1, HLA-DQA1 |
| | 6 | rs9443960 | 82119722 | T/C | intergenic | 51.7% | 52.3% | -0.026 (0.004) | 0.018 | 1.49*10^-8 | LINCO2542, IBTK |
| | 9 | rs10046912 | 84028173 | T/C | intergenic | 76.2% | 76.9% | -0.031 (0.006) | 0.013 | 1.67*10^-8 | RMI1 |
| | 14 | rs1245193 | 26449262 | T/C | intron | 83.1% | 82.5% | 0.034 (0.006) | 0.044 | 2.88*10^-8 | NOVA1 |
| | 18* | rs8088342 | 25030804 | T/G | intergenic | 63.9% | 63.2% | 0.026 (0.005) | 0.025 | 3.18*10^-8 | WBP2P1, ZNF521 |
| | 22 | rs9614155 | 30186009 | G/C | intron | 15.7% | 16.6% | -0.059 (0.006) | 0.074 | 2.77*10^-21 | HORMAD2, MTMR3, LIF |

Chr chromosome, Ref reference, Alt alternative, SE standard error, PAR population attributable risk.

Associations between each SNP and phenotypes were analyzed with REGENIE[44] using an additive model. The threshold for statistical significance was p < 5 * 10^-8.

*locus not credible according to SuSie finemapping.

**Table 3 | Lead SNPs of the loci associated with sub-phenotypes Pulpitis and Necrosis of pulp and apical periodontitis in the FinnGen discovery cohort**

| Phenotype | Chr | rsID | Position (GRCh38) | Ref/alt allele | Type | Alt allele frequency | | Beta (SE) | PAR | p-value | Nearest genes |
|---|---|---|---|---|---|---|---|---|---|---|---|
| | | | | | | Cases | Controls | | | | |
| Pulpitis | 1 | rs2997084 | 65928049 | C/G | intron | 37.5% | 38.6% | -0.043 (0.007) | 0.052 | 2.99*10^-9 | PDE4B |
| | 6 | rs9270664 | 32598372 | G/A | regulatory region | 52.3% | 51.2% | 0.049 (0.007) | 0.038 | 5.38*10^-12 | HLA-DRB1, HLA-DQA1 |
| | 6 | rs9449423 | 82117693 | T/C | intergenic | 51.3% | 52.3% | -0.043 (0.007) | 0.035 | 1.03*10^-9 | LINCO2542, IBTK |
| | 9 | rs1417738 | 99807317 | A/G | intron | 32.7% | 33.7% | -0.041 (0.007) | 0.052 | 3.04*10^-8 | NR4A3 |
| | 14 | rs1245193 | 26449262 | T/C | intron | 83.4% | 82.6% | 0.053 (0.009) | 0.079 | 1.33*10^-8 | NOVA1 |
| | 17 | rs1292071 | 59849550 | T/C | upstream | 47.1% | 46.1% | 0.038 (0.007) | 0.032 | 4.62*10^-8 | MIR11, TUBD1, VMP1 |
| | 18 | rs8088342 | 25030804 | T/G | intergenic | 64.2% | 63.2% | 0.042 (0.007) | 0.045 | 5.20*10^-9 | WBP2P1, ZNF521 |
| | 22 | rs9614152 | 30167354 | G/A | intron | 15.2% | 16.5% | -0.093 (0.009) | 0.141 | 2.81*10^-22 | HORMAD2, MTMR3, LIF |
| Necrosis of pulp and apical periodontitis | 2 | rs36044739 | 141650792 | CT/C | intron | 5.67% | 5.36% | 0.063 (0.011) | 0.005 | 1.26*10^-8 | LRPIB |
| | 3 | rs62253088 | 85351651 | T/C | intron | 71.5% | 72.1% | -0.031 (0.006) | 0.012 | 2.47*10^-8 | CADM2, MIR5688 |
| | 4 | rs79409042 | 106046283 | T/C | 3-prime UTR | 4.25% | 3.94% | 0.074 (0.013) | 0.005 | 6.19*10^-9 | TBCK |
| | 5* | rs6596024 | 131434092 | C/A | intron | 34.8% | 35.4% | -0.029 (0.005) | 0.023 | 4.94*10^-8 | RAPGEF6, CDC42SE2 |
| | 6 | rs9274719 | 32669693 | T/C | upstream gene | 23.3% | 22.6% | 0.036 (0.006) | 0.013 | 1.14*10^-9 | HLA-DQB1, HLA-DQA1 |
| | 8 | rs35858015 | 6328033 | C/A | regulatory region | 4.08% | 3.77% | 0.074 (0.013) | 0.005 | 1.46*10^-8 | MCPH1 |
| | 22 | rs9614155 | 30186009 | G/C | intron | 15.6% | 16.6% | -0.064 (0.007) | 0.083 | 2.23*10^-20 | HORMAD2, MTMR3, LIF |

Chr chromosome, Ref reference, Alt alternative, SE standard error, PAR population attributable risk.

Associations between each SNP and phenotypes were analyzed with REGENIE[44] using an additive model. The threshold for statistical significance was p < 5 * 10^-8.

*locus not credible according to SuSie finemapping.

regions spanned 208–343 kb containing the *HLA-DRA/B* and *HLA-DQA/B* genes in chromosome 6 (Supplementary Figs. 2–4). The lead SNP (rs9270664) of the phenotype Pulpitis had the strongest association ($p = 5.38 * 10^{-12}$) and PAR of 0.049. The lead SNPs of the phenotypes Pulpal and apical diseases and Pulpitis were located between *HLA-DRB1* and *HLA-DQA1* in the middle of the associated region, while the lead SNP of the phenotype Necrosis of pulp or apical periodontitis was located downstream of the most centromeric gene in the associated region, *HLA-DQB1-AS1*.

The discovery GWAS also identified several other associated loci (Table 2), including three in chromosomes 6 (near *IBTK*), 14 (near *NOVA1*), and 18 (near *ZNF521*) shared by the phenotypes Pulpitis and Pulpal and apical diseases. For the phenotype Pulpal and apical diseases, two other loci in chromosomes 2 (near *BCL11A*) and 9 (near *RMI1*) (Table 3), and for the phenotype Pulpitis, three other loci in chromosomes 1 (near *PDE4B*), 9 (near *NR4A3*), and 17 (near *VMP1* and *TUBD1*) (Table 3) were identified. For the phenotype Necrosis of pulp or apical periodontitis, we identified additional associations in chromosomes 2, 3, 4, 5 and 8. Their lead SNPs were near genes *LRP1B*, *CAMD2*, *TBCK*, *RAPGEF6*, and *MCPH1*, respectively. Most of these loci contained several significant and credible variants (Tables 2 and 3).

## Association of endodontic infections with several lead SNPs were replicable

Register and genotype data deriving from two FinnGen sub-cohorts, COROGENE[13] and NFBC1966/1986[14,15], were used for replication of the results obtained from the discovery GWASs (FinnGen-replication, Replication study 1). In total, these cohorts included 14,705 individuals, whose characteristics are presented in Table 1. From the GWAS of phenotype Pulpitis, lead SNPs rs9270664, rs1245193, and rs8088342 displayed significant associations with Pulpal and apical diseases, Pulpitis or Necrosis of pulp or apical periodontitis (Supplementary Table 1) in the Replication study 1.

Replication 2 with the Estonian Biobank (EstBB)[16] register data with corresponding phenotypes showed significant associations for rs9449423, rs9443960, and rs6596024. The associations with two lead SNPs near the *HORMAD2* gene (rs9614152 and rs9614155) were replicated in both FinnGen-replication and EstBB cohorts. The association of one additional lead SNP, rs9270911, was replicated in the NFBC cohort (replication 3) with phenotype apical lesions (Supplementary Table 1). From a total of 22 associations identified in the discovery GWAS, we were able to replicate 13 associations, and the associations with the SNPs near the *HORMAD2* gene in chromosome 22 were the strongest ones in the replication cohorts.

## Replication of associations from previous studies

Among 18 SNPs analysed (Supplementary Table 2), one presented a significant association with the phenotype Pulpitis (rs12800372, $p = 0.002$, near genes *TPCN2 / LOC338694*), whereas no significant associations with Pulpal and apical diseases or Necrosis of pulp or apical periodontitis were observed.

## Considering caries as a confounder

To investigate the role of caries in the pathogenesis of endodontic infections, we re-analyzed the associations of lead SNPs with our three phenotypes adjusting for the DMFS index (Supplementary Data 2). Among the lead SNPs for the phenotype Pulpal and apical diseases, rs143351662 (*BCL11A*), rs9443960 (*IBTK*), rs10046912 (*RMI1*), and rs9614155 (*HORMAD2*) retained their significance when adjusted for caries. Among the other two phenotypes, only rs1245193 (*NOVA1*) and rs62253088 (*CAMD2*) lost the significance. Furthermore, the association of our lead SNPs with caries was analysed using two caries phenotypes based on ICD-10 codes, dental caries (ICD-10 category K02) and caries of dentine (ICD-10 code K02.1) (Supplementary Data 3). Variations in chromosome 2 near *LRP1B*, chromosome 3 near *CAMD2*,

chromosome 6 with genes *HLA-DRB1*, *-DQB1*, and *-DOA1*, chromosome 9 near *NR4A3*, chromosome 14 near *NOVA1*, and chromosome 18 near *ZNF51* associated significantly with caries phenotypes.

## Differences between males and females

Results from sex-specific GWAS analyses are listed in Supplementary Data 2. The associations were consistent in males and females. Directions of associations were similar in males and females, even though there were some differences in the significance levels.

## Genes within the associated regions are expressed in dental pulp tissue and regulated by the lead SNPs

To provide insights into the expression and role of the associated genes and variants, we performed further analyses, which are summarized in Supplementary Data 4 and 5 and Supplementary Table 3. Interrogation of the Gene Expression Omnibus (GEO) database indicated various levels of expression of genes in the associated regions or in their close neighbourhood in the dental pulp (Supplementary Data 4). The normalized expression varied from high (multiple times of mean of global expression) to relatively low. Importantly, several genes in chromosome 22 and in the HLA-region were strongly expressed in dental pulp tissue. Moreover, analysis also suggested upregulation of several of these genes in pulpitis.

Analysis of the variants indicated only a few cases in which the variants affected the protein sequence (Supplementary Data 1). Two significant missense variants and several UTR single nucleotide variants were identified in *HLA-DQA1*, *HLA-DQB1*, and *HLA-DQB1-AS1*. The most credible 3-UTR candidate variants were found in *TBCK* (rs79409042, chromosome 4) in the phenotype Necrosis of pulp or apical periodontitis, in *HNRNPK* (rs696825, chromosome 9) in Pulpal and apical diseases, and in *MTMR3* (rs9983, chromosome 22) in Pulpitis. We predicted the regulatory significance of selected variants in each locus using Haploreg and RegulomeDB databases (Supplementary Data 5). In most loci, several or at least some of the most credible variants were predicted to have regulatory relevance, either by rank of at least 1 f by RegulomeDB, by affecting several transcription factor binding sites, or by residing in a region of enhancer activity (Supplementary Data 5). Among the 15 independent loci, 11 of the 19 lead SNPs were classified into RegulomeDB rank 1. Of note, the lead SNPs and several other most credible variants in the chromosome 22 and HLA loci as well as in loci in chromosomes 3, 6, 9, 14 and 17 carried multiple indications of regulatory relevance, thus likely affecting transcriptional regulation and linked to the expression of gene targets.

*Cis*-eQTLGen identified 27 genes affected in blood cells by our lead SNPs, whereas pQTL using two different proteomics analyses displayed significant associations with nine plasma proteins (Supplementary Table 3). According to the eQTL data summarized in the Fivex database, the lead SNPs in chromosome 22 had high probability in multiple tissues to affect the expression of the *MTMR3* gene, and lower probability to regulate the expressions of *LIF*, *HORMAD2*, and other genes. HLA genes were regulated by the lead SNPs near the *HLA-DRB1* and *HLA-DQB1* genes in multiple tissue types. No *trans*-eQTL were observed.

The protein coding genes displayed similar GO features both in Pulpitis and Necrosis of pulp or apical periodontitis (Supplementary Table 4). The most significant GO-terms were Peptide antigen assembly with MHC class II protein complex, MHC class II receptor activity, MHC class II protein complex binding, Peptide antigen binding, Luminal side of endoplasmic reticulum membrane, and ER to Golgi transport vesicle membrane with $p$-values $< 10^{-11}$.

## Endodontic infections are independently associated with HLA allele frequencies

As the GWAS results showed associations with genetic variation in the HLA region, especially near genes *HLA-DRB1* and *HLA-DQB1*, we further investigated the HLA alleles in the FinnGen discovery cohort (Table 4).

**Table 4 | Significant associations of HLA alleles with endodontic infections in FinnGen**

| HLA gene | Allele | Freq | Phenotype | | | | | | |
|---|---|---|---|---|---|---|---|---|
| | | | **Pulpal and apical diseases** | | **Pulpitis** | | **Necrosis of pulp or apical periodontitis** | |
| | | | Beta (SE) | FDR | Beta (SE) | FDR | Beta (SE) | FDR |
| A | 02:01 | 0.30 | 0.020 (0.005) | **0.007** | 0.026 (0.008) | **0.048** | 0.020 (0.006) | **0.03** |
| | 03:01 | 0.22 | -0.020 (0.006) | **0.02** | -0.027 (0.008) | 0.07 | -0.018 (0.007) | 0.07 |
| | 32:01 | 0.04 | -0.038 (0.013) | **0.049** | -0.052 (0.020) | 0.19 | -0.033 (0.014) | 0.17 |
| C | 01:02 | 0.07 | 0.029 (0.009) | **0.04** | 0.013 (0.070) | 0.39 | 0.030 (0.010) | 0.06 |
| DRB4 | 01:03 | 0.22 | 0.027 (0.006) | **0.0002** | 0.041 (0.009) | **0.0002** | 0.030 (0.006) | **0.0001** |
| DRB1 | 01:01 | 0.18 | -0.018 (0.006) | **0.04** | -0.027 (0.009) | 0.08 | -0.019 (0.007) | 0.07 |
| | 04:01 | 0.09 | 0.045 (0.009) | **3.05*10⁻⁵ᐟ** | 0.063 (0.013) | **0.0002** | 0.049 (0.009) | **0.04** |
| DQA1 | 01:01 | 0.19 | -0.018 (0.006) | **0.04** | -0.025 (0.009) | 0.09 | -0.018 (0.007) | 0.07 |
| | 03:01 | 0.11 | 0.030 (0.008) | **0.003** | 0.048 (0.012) | **0.002** | 0.036 (0.008) | **0.001** |
| | 03:03 | 0.02 | 0.072 (0.020) | **0.007** | 0.031 (0.030) | 0.64 | 0.081 (0.021) | **0.005** |
| | 05:05 | 0.07 | 0.028 (0.010) | **0.049** | 0.034 (0.015) | 0.15 | 0.028 (0.011) | 0.09 |
| DQB1 | 03:01 | 0.11 | 0.032 (0.008) | **0.002ᐟ** | 0.028 (0.012) | 0.13 | 0.032 (0.008) | **0.005ᐟ** |
| | 03:02 | 0.12 | 0.028 (0.007) | **0.003** | 0.045 (0.011) | **0.002** | 0.033 (0.008) | **0.001** |
| | 05:01 | 0.19 | -0.017 (0.006) | **0.04** | -0.027 (0.009) | 0.07 | -0.018 (0.006) | 0.07 |

Additive models adjusted for age, imputed sex, and genetic principal components 1–10. Statistically significant FDRs are bolded. FDR < 0.05 was considered statistically significant.
ᐟ Significant also in a model adjusted for age, imputed sex, autoimmune diseases, and genetic PCs 1–10.
Discovery cohort with 13,2124 cases and 35,3106 controls (Pulpal and apical diseases); 48,120 cases and 35,3106 controls (Pulpitis); 103,832 cases and 353,106 controls (Necrosis of pulp or apical periodontitis).
Freq, frequency; SE, standard error; FDR, false discovery rate

Pulpal and apical diseases were significantly associated with DRB1 * 01:01, DRB1 * 04:01, DQB1 * 03:01, DQB1 * 03:02, and DQB1*05:01. Additionally, associations were observed with A * 02:01, A * 03:01, A * 32:01, C * 01:02, DRB4 * 01:03, DQA1 * 01:01, DQA1 * 03:01, DQA1 * 03:03, and DQA1 * 05:05 among the 187 alleles investigated. Associations between endodontic infections and DRB1*04:01 and DQB1 * 03:01 were independent of autoimmune diseases.

In the FinnGen replication cohort, we were able to validate the associations of endodontic infections with HLA alleles A * 32:01, DRB4 * 01:03, DQA1 * 03:01, and DQB1 * 03:02. These associations were independent of autoimmune diseases (Supplementary Tables 5 and 6). Among these, HLA-A, DQA1, and DQB1 are in strong LD forming a haplotype, which was further observed in the FinnGen-replication cohort, displaying significant associations between HLA alleles A * 32:01, DRB4 * 01:03, DQA1 * 03:01, and DQB1 * 03:02 and endodontic infections.

Despite the association of the phenotype Pulpal and apical diseases with some HLA-A and HLA-C alleles, no associations were detected between endodontic infections and KIR gene content (Table 5). The phenotype Pulpitis displayed positive associations with 2DL5, 2DS5, and 3DS1. The associations were independent of autoimmune diseases, but they could not be replicated in the FinnGen-replication cohort (Supplementary Table 7).

### The lead SNPs associate with several phenotypes in FinnGen
The lead SNPs identified in the discovery GWAS were associated with multiple other phenotypes in FinnGen (Supplementary Table 8). Notably, numerous diseases of autoimmune character were detected. In addition, especially the lead SNP rs6596024 near *RAPGEF6* was associated with CVD phenotypes, such as coronary revascularization, angina pectoris, and ischaemic heart disease.

### Endodontic infections display genetic correlations with cardiovascular phenotypes
Narrow sense heritability ($h^2$) of our three phenotypes varied between 0.017 and 0.021 (SE 0.001) (Supplementary Table 9, Supplementary Fig 5). For the other FinnGen phenotypes analysed, $h^2$ varied between 0.0016 and 0.2823, and the highest $h^2$ values were observed for BMI, current smoking status, and caries. Genetic correlations between the phenotypes Pulpal and apical diseases, Pulpitis, and Necrosis of pulp or apical periodontitis were high ($r_g = 0.96$–$0.99$), as expected. When analysing genetic correlations between Pulpal and apical diseases and other FinnGen phenotypes, the strongest correlations were observed for pain ($r_g = 0.63$, $p = 5.9 * 10^{-120}$, Fig. 2 and Supplementary Table 10) and caries ($r_g = 0.54$, $p = 8.5 * 10^{-34}$). In addition, strong correlations were detected between endodontic infections and cardiovascular risk factors (smoking status, $r_g = 0.44$, $p = 1.3 * 10^{-46}$; BMI, $r_g = 0.4$, $p = 5.9 * 10^{-60}$; type 2 diabetes, $r_g = 0.29$, $p = 7.2 * 10^{-23}$) and also with cardiovascular phenotypes (CVD, $r_g = 0.39$, $p = 1.1 * 10^{-47}$; stroke, $r_g = 0.32$, $p = 2.7 * 10^{-13}$).

## Discussion
We identified 15 independent loci associating with endodontic infections in our genome-wide association analyses of 485,230 Finnish individuals from the FinnGen cohort using national register data. Twelve lead SNPs were associated with pulpal and apical diseases independently of caries, whereas the association of three SNPs was dependent on caries and three were associated with both endodontic infections and caries. In our functionally oriented analyses, genes near our lead SNPs were expressed in the dental pulp, and several lead SNPs participated in the regulation of multiple genes in various tissues, thus suggesting their effect on gene expression. In addition, the GWAS revealed an association between endodontic infections and genetic variation in the HLA region. Further analyses using imputed HLA allele frequencies indicated that especially HLA genes *HLA-A*, *HLA-DRB1*, *HLA-DRB4*, *HLA-DQA1*, and *HLA-DQB1* may play a role in endodontic infections independent of autoimmune diseases. Genetic correlations were found between endodontic infections and several pain, autoimmune, cardiometabolic, and cardiovascular phenotypes. Our genome-wide study suggests that genetic variation contributing to immune dysregulation is involved in the pathogenesis of endodontic infections, which have considerable genetic similarity with other complex traits.

All three phenotypes, Pulpal and apical diseases, Pulpitis, and Necrosis of pulp or apical periodontitis, showed the strongest association with variants near *HORMAD2* (HORMA domain containing 2) gene in chromosome 22 independently of caries, suggesting that this

**Table 5 | Associations of KIR gene contents with endodontic infections in FinnGen**

| | KIR gene | Freq | Phenotype | | | | | | |
|---|---|---|---|---|---|---|---|---|---|
| | | | Pulpal and apical diseases | | Pulpitis | | Necrosis of pulp or apical periodontitis | | |
| | | | Beta (SE) | FDR | Beta (SE) | FDR | Beta (SE) | FDR |
| Inhibitory | **2DL1** | 0.51 | 0.008 (0.025) | 0.80 | 0.017 (0.037) | 0.79 | -0.004 (0.027) | 0.98 |
| | **2DL2** | 0.79 | -0.009 (0.007) | 0.62 | 0.003 (0.010) | 0.83 | -0.016 (0.007) | 0.19 |
| | **2DL3** | 0.53 | -0.001 (0.014) | 0.92 | -0.024 (0.020) | 0.38 | 0.004 (0.014) | 0.98 |
| | **2DL5** | 0.74 | 0.002 (0.007) | 0.80 | 0.030 (0.010) | **0.03**[*] | -0.007 (0.007) | 0.84 |
| | **3DL1** | 0.53 | -0.009 (0.013) | 0.76 | -0.027 (0.020) | 0.35 | -0.004 (0.014) | 0.98 |
| Activating | **3DS1** | 0.78 | 0.007 0.007) | 0.74 | 0.029 (0.010) | **0.03**[*] | 0.001 (0.008) | 0.98 |
| | **2DS1** | 0.78 | 0.013 (0.007) | 0.61 | 0.023 (0.010) | 0.06[*] | 0.009 (0.007) | 0.63 |
| | **2DS2** | 0.79 | -0.009 (0.007) | 0.62 | 0.002 (0.010) | 0.83 | -0.015 (0.007) | 0.19 |
| | **2DS3** | 0.87 | -0.007 (0.007) | 0.76 | 0.014 (0.012) | 0.38 | -0.015 (0.009) | 0.27 |
| | **2DS4** | 0.53 | -0.009 (0.013) | 0.76 | -0.027 (0.020) | 0.35 | -0.003 (0.014) | 0.98 |
| | **2DS5** | 0.82 | 0.008 (0.007) | 0.71 | 0.027 (0.011) | **0.05**[*] | 0.003 (0.008) | 0.98 |
| Pseudogene | **2DP1** | 0.51 | 0.013 (0.025) | 0.80 | 0.029 (0.038) | 0.59 | -0.001 (0.028) | 0.98 |

Additive model adjusted for age, sex, and genetic principal components 1–10. Statistically significant FDRs are bolded. FDR < 0.05 was considered statistically significant.
[*] Significant also in a model adjusted for age, sex, autoimmune diseases, and genetic principal components 1–10.
Freq, frequency; SE, standard error; FDR, false discovery rate

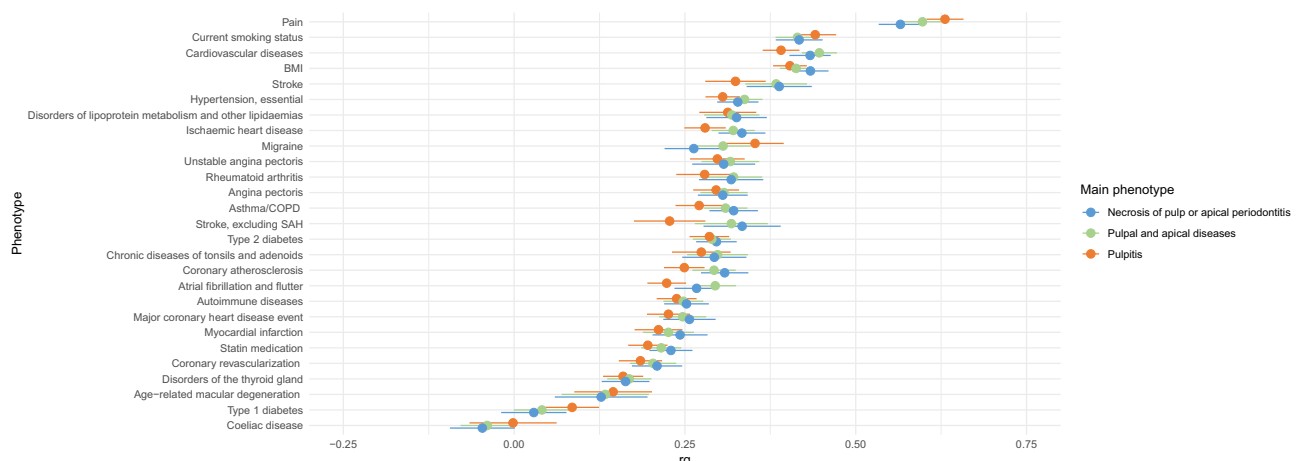

**Fig. 2 | Genetic correlations.** We assessed genetic correlation between endodontic infections and other FinnGen clinical phenotypes using linkage disequilibrium score regression. Pairwise genetic correlations ($r_g$) between the phenotypes and their heritability estimates were used to quantify the shared genetic variance relative to the square root of their respective SNP heritability estimates. In total, 31 phenotypes were analysed and assessed for correlation with 3 endodontic phenotypes. Markers indicate the estimated magnitude of genetic correlation ($r_g$) and error bars represent standard errors.

genetic locus is associated especially with progression of deep caries lesions into endodontic infections. The associations of the lead SNPs rs9614152 and rs9614155 were replicated both in the FinnGen-replication cohort and EstBB. Lead SNPs displayed significant *cis*-eQTLs in genes *MTMR3, ASCC2, SF3A1, THOC5, NEFH*, and *DUSP18*, which all were expressed also in dental pulp tissue. Upregulation of *MTMR3* (myotubularin-related protein 3) was seen in multiple tissues, such as immune, adipose, oesophagus, and blood vessel tissues. eQTL studies also suggested a negative effect of the lead SNPs on the expression of *LIF. LIF* codes for leukaemia inhibitory factor, an IL6-family cytokine, and has been reported to aggravate pulpitis in mouse and experimental studies[17]. These earlier results support our observation that *LIF* is associated with endodontic infections and that its expression is upregulated in dental pulp during pulpitis. Polymorphisms within *MTMR3/HORMAD2/LIF/OSM* have previously been associated with tonsillectomy and inflammatory bowel diseases[18–20], suggesting that the underlying genes might have a role especially in mucosal immune homoeostasis. *MTMR3* was shown to modulate pattern recognition receptor (PRR)-induced cellular responses by inhibiting autophagy and enhancing pro-inflammatory signalling[20]. Similarly as in other infections, autophagy has multiple functions in endodontic infections, including direct elimination of microbes, maintaining energy supply during stress, starvation, and hypoxia of the dental pulp, and regulating inflammation, thus being causative in the development and maintenance of periapical lesions[21].

Genetic variations in Class II HLA genes were associated with endodontic infections. The lead SNPs were associated also with caries, suggesting that genetic variation in the HLA region may be involved in the initiation and progression of both diseases. HLA molecules of this class are almost solely expressed in professional antigen presenting cells activated by microbial infections and generating the humoral immune response[22]. In our gene ontology analyses of the protein-coding gene families, the results were dominated by MHC Class II -related pathways:Pulpal and apical diseases and Pulpitis were associated with genetic variants near *HLA-DRB1*, whereas Necrosis of pulp or apical periodontitis associated with variation near *HLA-DQA1*. These variants were associated with downregulation of *HLA-DRB1, HLA-DRB5, HLA-DQA1, HLA-DQB1*, and *HLA-DQB1-AS1*, and upregulation of *HLA-*

*DQA2* and HLA-*DQB2* in multiple tissues. Of these, HLA-*DRB1* had a particularly high expression level in the dental pulp. The HLA allele analyses emphasized the importance of A*32:01 (frequency 4%) as protective and DRB4*01:03 (22%), DRB1*04:01 (9%), DQA1*03:01 (11%), and DQB1*03:02 (12%) as risk alleles. Altogether, 14 HLA alleles were associated with the phenotypes in the discovery analyses and four of these associations could be replicated in other populations. Although in populations of European ancestry, haplotype HLA-DRB1*04-DQA1*03-DQB1*03:02 is associated with highest risk of type 1 diabetes[23], the associations of single alleles with endodontic infections were independent of autoimmune diseases – a combination of 45 autoimmune disease endpoints including type 1 diabetes. Associations between HLA classes and oral infectious diseases have not been studied widely. A meta-analysis has found an association between marginal periodontitis and HLA Class I antigens, but not with Class II antigens[24], whereas HLA alleles B*15, B*51, and DRB1*12 were later inversely associated with periodontal disease[25]. Previous studies have not found significant associations between HLA alleles and caries experience in adult populations[25,26]. Not surprisingly, our results show that the HLA system - which displays extensive genetic variation to enable its role in the regulation of the immune responses - plays a role also in endodontic infections, which derive from the dysbiotic oral microbiome and the consequent host defence mechanisms[1–3].

Killer cell immunoglobulin-like receptors (KIRs) have an essential role in the control of natural killer (NK) cell function[27]. Pulpitis was associated with the inhibitory KIR-2DL5 receptor gene, whose ligands are autologous HLA-class I molecules. Through interactions with their ligands, the inhibitory KIRs prevent NK cells from killing healthy cells, thereby contributing to the maintenance of the self/nonself discrimination. Pulpitis was associated also with activated 2DS5 and 3DS1, which have been shown to predispose to several diseases in immunogenetic studies[22]. Whether they play a role in endodontic infections requires further investigation, since the associations could not be replicated in our replication cohorts.

We found that endodontic infections are heritable at a moderate level with $h^2$ estimates of 0.017-0.021 (SE = 0.001) and thus falling between $h^2$ for stroke ($h^2 = 0.019$, SE = 0.002) and type 1 diabetes ($h^2 = 0.023$, SE = 0.014). Single lead SNPs associated strongly with phenotypes that are not usually discussed in relation to endodontic infections, such as coeliac disease, hypothyroidism, anaemia, atopic dermatitis, and chronic diseases of tonsils and adenoids. Therefore, we further analysed genetic correlations of endodontic infections with selected autoimmune diseases, pain phenotypes, and cardiometabolic phenotypes. We found strong associations both with cardiometabolic disorders and cardiovascular diseases. Endodontic medicine examines the interrelationship between pathological conditions of periapical tissues and systemic health: patients with systemic problems, such as diabetes, may present an exacerbated response to endodontic pathogens, whereas apical periodontitis may accelerate pathogenesis of cardiometabolic diseases[28]. According to current consensus, the etiopathogenic link between periodontitis and cardiometabolic disorders is the low-grade systemic inflammation sustained by the oral dysbiosis and asymptomatic, chronic oral infections[29]. However, not all studies support an association. For example, patients with marginal periodontitis have metabolic profiles typical for inflammation, whereas untreated endodontic infections do not display as clear systemic inflammatory characters[30]. Apical periodontitis risk score was associated with circulatory system, endocrine/metabolic, and hematopoietic pathways in an earlier phenome-wide association analysis[11]. Our present results of genetic correlations additionally support the hypothesis that common susceptibility may link endodontic infections with cardiometabolic traits.

We selected genetic variants from an earlier publication[11] for replication in our discovery GWAS, but only one (rs12800372 near *TPCN2*) of those showed a significant association with our Pulpitis phenotype. This may be due to differences between the phenotypes,

since the earlier study examined participants with deep caries and the cases suffered from apical periodontitis, whereas the controls of the present study were not selected based on caries status. However, our adjusted association analyses showed that most of the associations between our genetic loci and endodontic infections were independent of caries. Two thirds of the lead SNPs identified in our discovery GWAS were replicable in other cohorts, especially in the FinnGen-replication and EstBB, where we were able to construct identical phenotypes. A few associations could also be replicated in smaller clinical studies, NFBC1966 and NFBC1986, with different but closely related available phenotypes, deep caries with pain and apical periodontitis increasing the reliability of our findings.

The diagnosis of an endodontic infection is reached after a careful clinical and radiographic examination. Based on both subjective and objective findings, the diagnoses may vary from reversible pulpitis to irreversible pulpitis and pulp necrosis. However, the disease often progresses to apical periodontitis, which may also be asymptomatic. Thus, albeit asymptomatic apical periodontitis is highly prevalent comprising ~60% of the cases in the present study, it often is an incidental finding in oral radiographs. Despite strong genetic correlation, clearly, the two disease stages, Pulpitis and Necrosis of pulp or apical periodontitis, displayed different genetic profiles: only genetic variants in chromosome 22 and within the HLA region associated significantly with both phenotypes. The genes regulated by the lead SNPs of the phenotype Pulpitis had described functions in autophagy (*VMP1*), cilia function (*INVS* and *TUBD1*), nerve function (*NOVA1*), dentin formation, tooth pain transmission (*INVS*)[31], and pulpitis (*LIF*)[17], whereas *PDE4B* is related to sweet taste signalling and associated with the expression of leptin receptor involved in regulating the balance between food intake and energy consumption[32]. *PDE4B* (cAMP-specific 3′, 5′-cyclic phosphodiesterase 4B) is expressed in central nervous system, and it metabolites secondary messenger molecules. On a spinal level, it is involved in the modulation of chronic pain[33], and thus it might represent the link between pain phenotypes and endodontic infections observed in the present study. In turn, several lead SNPs of Necrosis of pulp or apical periodontitis may associate with tumour suppression (*LRP1*, *CAMD2*, *MCPH1*), while others are involved in plasma membrane structure, assembly, and transport. Gene ontology analyses revealed enrichment of a few pathways not seen for other phenotypes. These were Detection of bacterium, Humoral immune response, and Carnitine transport activities. Indeed, apical periodontitis is accompanied with a strong humoral immune response against oral pathobionts[34], and assorted bacteria can benefit from carnitine in both aerobic and anaerobic environment[35]. Ineffective immune response may obviously lead to the progression of endodontic infections.

One strength of the study is the very large sample size of almost 500,000 participants, including 353,106 controls. Thus, we were able to maintain large cohort sizes even when investigating endodontic disease sub-entities, Pulpitis and Necrosis of pulp or apical periodontitis. These two diagnoses represent different disease stages, which display both overlapping and different genetic risk profiles. Consequently, the large number of observations enabled us to identify 15 GWAS-significant loci mainly associated with the effectiveness of immune responses. Only one GWAS on apical periodontitis without significant associations has been published earlier[11]. The Finnish nationwide electronic health registers have been originally established principally for administrative purposes to monitor the usage of health care services of all residents. Their strength is the wide national utilization, but the limitation may be diagnostic challenges since endodontic infections may remain undiagnosed when they are asymptomatic, and therefore false negatives can arise. This especially concerns apical periodontitis, which has a global prevalence 52% at the individual level and 5% at the tooth level[3]. In FinnGen population, the prevalence was on a similar level in women, whereas men were diagnosed more seldom. It is known that Finnish men use dental

services less frequently than women[36], which may represent a recall bias. Socioeconomic status, which affects the use of dental services, was not considered here either. The main limitations of LD Score regression are sample size and heritability of the traits. The sample size of our study was sufficiently large for the analysis, but the absolute $h^2$ values for endodontic infections seemed rather moderate. This, however, was considered to originate from the use of observed scale where heritability is directly tied to the proportion of cases in the sample, and not liability scale adjusting for population prevalence, which is unknown. Heritability of endodontic infections was of the same magnitude as that of stroke or coronary heart disease, which both have strong genetic component and are considered 30-40% heritable[37].

We have identified moderate heritability and multiple genetic variants associated with endodontic infections, which are highly common multifactorial oral diseases among adults. The results suggest involvement of genes that play a role in mucosal immune homoeostasis, cellular responses to inflammatory stimuli, and antigen recognition and/or presentation.

## Methods

### FinnGen cohort
FinnGen[12] is a large-scale biobank initiative integrating genomic data with information from Finnish health registries. National health registries that record health events, such as ICD-based diagnosis codes, include the National Hospital Discharge Registry (inpatient and outpatient), Causes of Death Registry, the National Infectious Diseases Registry, Cancer Registry, Primary Health Care Registry (outpatient), and Medication Reimbursement Registry. The Finnish current care guidelines for endodontic infections provide instructions on when and how to use ICD-10 diagnosis codes in clinical situations. These codes are recorded by dentists at every dental visit and are stored in the health registers. FinnGen Release 12 (September 2023) included data from 520,210 participants. We defined individuals with pulpal or periapical infections by searching for ICD-10 category K04 diagnosis codes from the hospital ($n = 1018$), primary care ($n = 120,881$), and cause of death registries ($n = 3$). Diagnoses from the hospital discharge registry were available since year 1995 and the primary care registry data was available since 2011.

The frequencies of ICD-10 diagnosis codes for pulpal and apical diseases are presented in Supplementary Fig 1. Codes K04.01, K04.02, K04.03, K04.04, and K04.05 are not included in the Finnish ICD-10 code library, and thus, individuals with these codes were excluded from the statistical analyses. We used three phenotypes for the GWAS: 1) any diagnosis in category K04 named as phenotype Pulpal and apical diseases; 2) Pulpitis (K04.0, K04.00, K04.08, and K04.09); and 3) Necrosis of pulp or apical periodontitis (K04.1, K04.2, K04.3, K04.4, K04.5, K04.6, K04.60, K04.61, K04.62, K04.63, K04.69, and K04.7). Individuals without any K04 category diagnosis were used as controls for each of the three phenotypes.

DMFS (the number of decayed, missing, and filled surfaces) index was available for 84,690 individuals for the phenotype Pulpal and apical diseases (36,137 cases and 48,553 controls), 62,106 for the phenotype Pulpitis (13,552 cases and 48,553 controls), and 78,145 for the phenotype Necrosis or apical periodontitis (29,592 cases and 48,553 controls) from FinnGen Release 11.

The results deriving from the FinnGen discovery GWAS and using ICD10-codes for were replicated in a subset of FinnGen consisting of COROGENE[13] and Northern Finland Birth Cohorts (NFBC) 1966 and 1986[14,15] (Replication study 1), and in the Estonian Biobank[16] (Replication study 2). Additionally, lead SNPs were investigated in NFBC 1966 and 1986 (Replication study 3), which included detailed oral health data from clinical and radiographic examinations.

### Northern Finland birth cohorts 1966 and 1986
The Northern Finland Birth Cohorts 1966 and 1986 Studies are lifespan cohort studies focusing on individuals born in 1966 and 1986 in the two northernmost provinces of Finland[14,15]. Comprehensive oral health examinations were included in the NFBC 1966 study (1964 participants) in 2012-2013 and in the NFBC 1986 study (1807 participants) in 2019-2020, respectively[38,39].

Findings of deep caries (ICDAS codes 5-6), questionnaire data, and diagnosed apical lesions from panoramic x-rays were utilized to create two proxy phenotypes for pulpitis and pulpal pathology: i) Deep caries and regular pain: This phenotype combines deep caries findings with reported regular dental pain to indicate recent and regular pulpitis-related symptoms, and ii) Apical lesions: This phenotype uses the presence of apical lesions in panoramic x-rays.

Pain related to pulpitis was assessed through two questionnaire questions: Q1: "Was the reason for the previous dental visit pain?" (yes/no), Q2: "Have you felt any pain or discomfort in your mouth during the last month?" (with six answer options ranging from "very often" to "never"). A combined "regular pain" variable was created from these questions, where a "yes" to Q1 or "very often" or "fairly often" to Q2 indicated pain.

### Estonian biobank cohort
The Estonian Biobank (EstBB)[16] is a population-based biobank with around 200,000 participants. 83% of the samples were from individuals of Estonian origin, 14% from individuals with Russian origin, and 3% from other ethnicities. In this study, data freeze 2020v4 was used. All biobank participants have signed a broad informed consent form and information on ICD-10 codes is obtained via regular linking with the national Health Insurance Fund and other relevant databases, with majority of the electronic health records having been collected since 2004[16]. The three phenotypes examined were: 1) any diagnosis in ICD10 category K04 named as phenotype Pulpal and apical diseases; 2) Pulpitis (K04.0); and 3) Necrosis of pulp or apical periodontitis (K04.1, K04.2, K04.4, K04.5, K04.6, K04.7).

### Genotyping and imputation
FinnGen samples were genotyped with Illumina (Illumina Inc., San Diego, CA, USA) and Affymetrix (Thermo Fisher Scientific, Santa Clara, CA, USA) arrays (https://www.finngen.fi/en/node/1996). GWAS data was imputed against the Finnish population-specific Sequencing Initiative Suomi (SISu) v4 reference panel. In sample-wise quality control, individuals with ambiguous gender, high genotype missingness (>5%), excess heterozygosity (+-4SD), and non-Finnish ancestry were excluded. In variant-wise quality control, variants with high missingness (>2%), low HWE $p$-value (<$10^{-6}$) and low minor allele count (MAC < 3) were excluded. All EstBB participants have been genotyped at the Core Genotyping Lab of the Institute of Genomics, University of Tartu, using Illumina Global Screening Array v1.0 and v2.0. Individuals were excluded from the analysis if their call-rate was <95% or if sex defined based on heterozygosity of X chromosome did not match sex in phenotype data. Before imputation, variants were filtered by call-rate <95%, HWE $p$-value < $10^{-4}$ (autosomal variants only), and minor allele frequency <1%. Population specific imputation reference of 2297 WGS samples was used for imputation[40-43]. The NFBC 1966 samples were genotyped using Illumina HumanCNV370DUO Analysis BeadChip and the NFBC 1986 using Illumina HumanOmniExpressExome-8v1.2. Imputation was performed with HRC r1.1 reference panel. In NFBCs 1966 and 1986, SNPs with minor allele frequency <1%, missingness rate >5%, and HWE $p$-value < 0.0001 were excluded.

### Association analyses
To discover genetic associations with our three phenotypes, we performed GWASs in FinnGen with REGENIE software (v2.2.4) (https://rgcgithub.github.io/regenie/), using an additive model[44]. All analyses were adjusted for age, imputed sex, genotyping chip, first 10 genetic principal components (PCs), and genotyping batch. The COROGENE and NFBC 1966/1986 cohorts were removed from the discovery

analyses in FinnGen. Population attributable risk (PAR) was estimated from risk allele frequencies and risk ratios with the formula:

$$PAR = 1 - \frac{1}{(1-p)^2 + 2p(1-p)RRhet + p^2 RR\,hom} \qquad (1)$$

where $p$ is the population frequency of the risk allele and RRhet and RRhom are the risk ratios for heterozygotes and homozygotes as calculated from the allele frequencies assuming Hardy-Weinberg equilibrium among cases and controls[45].

We replicated the associations between the phenotypes and the lead SNPs from the discovery analyses in a subset of FinnGen (Replication study 1) consisting of COROGENE and NFBC 1966 and 1986 cohorts by using REGENIE and similar adjustments as in the discovery analyses. To examine the associations of the top variants with other clinical endpoints, we used the summary statistics of the FinnGen R12 clinical endpoints (https://www.finngen.fi/en/researchers/clinical-endpoints). In the Estonian Biobank, the associations between SNPs and three phenotypes (Pulpal and apical diseases, Pulpitis, and Necrosis of pulp or apical periodontitis) were analyzed using REGENIE v2.2.4 with sex, age, age squared, and ten PCs as covariates (Replication study 2). In the NFBC 1966/1986 (Replication study 3) cohorts, the associations between the top SNPs and two endodontic phenotypes were analyzed by additive regression models adjusted for sex, age, batch, and ten PCs using SNPTEST v2.5.4 software.

We attempted to replicate the associations of the SNPs identified in a recent GWAS on apical periodontitis[11] using our summary statistics of the discovery GWASs.

### Analyses for the effect of caries
We studied the associations between of our lead SNPs and two caries phenotypes in FinnGen Release 11. These phenotypes were based on ICD10 codes: 1) cases: individuals with ICD10 codes in category K02 in any register, controls: those with no K02 category codes in any register; 2) cases: individuals with ICD10 code K02.1 (Caries of dentine) in any register, controls: those with no ICD10-code K02.1 in any register. We searched for our lead SNPs from the summary statistics of GWAS analyses for these caries phenotypes.

We additionally analysed the associations between our lead SNPs and phenotypes Pulpal and apical diseases, Pulpitis, and Necrosis of pulp or apical periodontitis in FinnGen with REGENIE using an additive model adjusted for age, imputed sex, genotyping chip, first 10 genetic PCs, genotyping batch, and DMFS.

### Finemapping
All genome-wide significant loci were further fine-mapped with SuSiE[46]. The fine-mapped regions extended 1.5 Mb upstream and downstream from the lead variant. The major histocompatibility complex (MHC) region in chromosome 6 was excluded from finemapping because of its complex LD structure. We computed in-sample linkage disequilibrium using LDStore2. Details of the finemapping pipeline are available in https://github.com/FINNGEN/finemapping-pipeline.

### Imputation of HLA alleles and KIR gene content and association analyses
Classical HLA alleles at four-digit resolution were imputed on the MHC SNP data as part of FinnGen processing pipeline using a Finnish reference panel[47] and HIBAG[48] algorithm. In both cohorts, KIR gene content at the level of absence/presence were imputed using a Finnish reference panel[49]. In FinnGen discovery cohort, the associations between the HLA alleles and KIR gene contents and three phenotypes (Pulpal and apical diseases, Pulpitis, and Necrosis of pulp or apical periodontitis) were calculated with REGENIE (v3.0.1) adjusted for age, imputed sex, and first 10 genetic PCs.

### In silico analyses
Gene expression in dental pulp was interrogated in the Gene Expression Omnibus (GEO; https://www.ncbi.nlm.nih.gov/geo/). Several studies were found which described RNAseq results from human or mouse pulp or in developing dental tissues. While the relative expression levels of different genes appeared largely similar in different studies of human adult dental pulp cells, we selected two studies[50,51] because they contained several replicates and TPM normalized data. The other also presented data from carious and pulpitis teeth. Gene expression differences between normal and affected pulp were also compared to data from experimental caries in mice[52] and from developing mouse tooth[53]. TPM normalized counts for individual genes in or near the associated regions were divided by mean TPM counts and a mean of these relative values from the two human studies were used to categorize the expression levels as presented in the Supplementary Data 4. The raw counts from mouse developing embryonic day 16 molar tooth germs were also compared to the raw counts from studies of human adult dental pulp.

The effects of the lead SNPs on the expression of genes were examined utilizing expression quantitative trait loci (eQTL) database browsers FIVEx (https://fivex.sph.umich.edu/)[54] and eQTLGen Phase I (https://www.eqtlgen.org/phase1.html)[55]. In the FIVEx, -log10 $p$-value > 7, and in eQTLGen Phase I, false detection rate (FDR) < 0.05 were considered statistically significant. Functional context of the top SNPs was annotated using Regulome DB database (https://www.regulomedb.org/regulome-search)[56] and Haploreg (https://pubs.broadinstitute.org/mammals/haploreg/haploreg.php).

Protein quantitative loci (pQTL) estimates for the FinnGen R12 were performed as a part of the FinnGen analysis pipeline. In brief, plasma protein levels were analysed with multiplex antibody-based immunoassay and multiplex aptamer-based immunoassay, Olink (https://olink.com) and SomaScan[57], respectively. Association tests between SNPs and protein levels were run using PLINK[58]. To identify significant pQTLs to our GWAS lead SNPs, we extracted the association results for the set of fine-mapped significant GWAS variants from the plasma proteomics summary statistics data. Proteins having a pQTL association with FDR < 0.05 were considered statistically significant.

Genes showing significant effects in eQTL and pQTL analyses were combined to Gene Ontology (GO) knowledgebase analyses (https://geneontology.org/)[59]. GO enrichment analysis was performed for biological processes, molecular functions, and cellular components. Results with FDR < 0.05 were considered statistically significant.

### Linkage disequilibrium score regression
We assessed genetic correlations between our three phenotypes and selected FinnGen clinical phenotypes using linkage disequilibrium score regression (LDSC, https://github.com/bulik/ldsc)[60]. LDSC software was used to calculate pairwise genetic correlations ($r_g$) between the phenotypes and their narrow sense heritability. In total, 31 phenotypes were analysed and assessed for correlation with 3 endodontic phenotypes, producing a total $n$ of 93 comparisons. After exclusion of 3 self-comparisons, 90 comparisons remained and the level of statistical significance after Bonferroni multiple comparison correction was predetermined as $p = 0.05/90 = 0.00056$.

### Reporting summary
Further information on research design is available in the Nature Portfolio Reporting Summary linked to this article.

## Data availability
The GWAS summary statistics generated in this study are available in the FinnGen public cloud bucket (https://storage.googleapis.com/fg-publication-green-public/F_2023_050_20250522/summary_statistics_pulpal_and_apical_diseases.zip). The individual-level data from FinnGen are available under restricted access due to the sensitive nature of the genotype and phenotype information. Individual-level genotypes

and register data from FinnGen participants can be applied via the Fingenious® services (https://site.fingenious.fi/en/) hosted by the Finnish Biobank Cooperative FinBB (https://finbb.fi/en/). Finnish Health register data can be applied by approved researchers via Finnish Data Authority Findata (https://findata.fi/en/data/).

## Code availability

We used publicly available software in our analyses: REGENIE (v2.2.4) https://rgcgithub.github.io/regenie/, PLINK (v2.0) https://www.cog-genomics.org/plink/2.0/, SNPtest (v2.5.4) https://www.chg.ox.ac.uk/~gav/snptest/, SuSiE https://github.com/stephenslab/susieR, and LDSC (v1.0.1) https://github.com/bulik/ldsc. A detailed description of data production and analysis including code used to run analyses is available in https://finngen.gitbook.io/documentation/. Please see https://github.com/FINNGEN/ for further code repositories used to run analyses in FinnGen.

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

## Acknowledgements

We want to acknowledge the participants and investigators of the FinnGen study. A full list of FinnGen Consortium members is shown in the Supplementary Data 6. The FinnGen project is funded by two grants from Business Finland (HUS 4685/31/2016 and UH 4386/31/2016) and the following industry partners: AbbVie Inc., AstraZeneca UK Ltd, Biogen MA Inc., Bristol Myers Squibb Inc. (and Celgene Corporation & Celgene International II Sàrl), Genentech Inc., Merck Sharp & Dohme LCC, Pfizer Inc., GlaxoSmithKline Intellectual Property Development Ltd., Sanofi US Services Inc., Maze Therapeutics Inc., Johnson&Johnson Innovative Medicine Inc., Novartis AG, Boehringer Ingelheim International GmbH and Bayer AG. Following biobanks are acknowledged for delivering biobank samples to FinnGen: Auria Biobank (www.auria.fi/biopankki), THL Biobank (www.thl.fi/biobank), Helsinki Biobank (www. helsinginbiopankki.fi), Biobank Borealis of Northern Finland (https:// www.ppshp.fi/Tutkimus-ja-opetus/Biopankki/Pages/Biobank-Borealis-briefly-in-English.aspx), Finnish Clinical Biobank Tampere (www.tays.fi/ en-US/Research_and_development/Finnish_Clinical_Biobank_Tampere), Biobank of Eastern Finland (www.ita-suomenbiopankki.fi/en), Central Finland Biobank (www.ksshp.fi/fi-FI/Potilaalle/Biopankki), Finnish Red Cross Blood Service Biobank (www.veripalvelu.fi/verenluovutus/ biopankkitoiminta), Terveystalo Biobank (www.terveystalo.com/fi/ Yritystietoa/Terveystalo-Biopankki/Biopankki/) and Arctic Biobank (https://www.oulu.fi/en/university/faculties-and-units/faculty-medicine/northern-finland-birth-cohorts-and-arctic-biobank). All Finnish Biobanks are members of BBMRI.fi infrastructure (https://www. bbmri-eric.eu/national-nodes/finland/). Finnish Biobank Cooperative FINBB is the coordinator of BBMRI-ERIC operations in Finland. The Finnish biobank data can be accessed through the Fingenious® services (https://site.fingenious.fi/en/) managed by FINBB. We want to acknowledge the participants of the Estonian Biobank for their contributions. The Estonian Genome Center analyses were partially carried out in the High Performance Computing Center, University of Tartu. We acknowledge the Estonian Biobank Research Team (Andres Metspalu, Mari Nelis, Lili Milani, Reedik Mägi, Georgi Hudjashov, Tõnu Esko; estb-bresearch@ut.ee) for data collection, genotyping, QC, and imputation. NFBC1966 46 year follow-up study received financial support from University of Oulu Grant no. 24000692, Oulu University Hospital Grant no. 24301140, and ERDF European Regional Development Fund Grant no. 539/2010 A31592. NFBC1986 33–35 year follow-up study received financial support from University of Oulu (Strategic funding from donations) and Oulu University Hospital (K65760). The oral health study was supported in part by the Research Council of Finland (former Academy of Finland, #326189). The work was supported by the Research Council of Finland (#340750 to P.J.P. and #326189 to J.M.L. and V.V.), Sigrid Juselius foundation (P.J.P.), Novo Nordisk fonden (P.J.P.), Paulo foundation (P.J.P.), Selma and Maja-Lisa Selander foundation (A.S. and P.J.P.), HUS Helsinki University Hospital Research Foundation (#Y2519SU008 to D.P.R.), and the Estonian Research Council (#PRG1291 to K.R., A.R., and P.P.). Open access was funded by Helsinki University Library.

## Author contributions

A.S.: conception and design of the study, analysis and interpretation of the data, drafting the manuscript K.H.: conception of the study, analysis and interpretation of the data, critically reviewing the manuscript J.R.: analysis and interpretation of the data, critically reviewing the manuscript J.M.L.: analysis and interpretation of the data, critically reviewing the manuscript U.P.: design of the study, critically reviewing the manuscript V.V.: analysis of the data, critically reviewing the manuscript O.K.: analysis and interpretation of the data, critically reviewing the manuscript FinnGen and Estonian Biobank Research Team: conception and design of the study, data acquisition K.R.: analysis of the data, critically

reviewing the manuscript A.R.: analysis of the data, critically reviewing the manuscript P.P.: analysis of the data, critically reviewing the manuscript M.P.: conception and design of the study, critically reviewing the manuscript J.S.: data acquisition, critically reviewing the manuscript A.S.H.: data acquisition, critically reviewing the manuscript P.M.: data acquisition, critically reviewing the manuscript U.K.G.: data acquisition, critically reviewing the manuscript A.L.S.: data acquisition, critically reviewing the manuscript D.P.R.: data acquisition, critically reviewing the manuscript V.A.: data acquisition, design of the study, critically reviewing the manuscript P.N.: analysis and interpretation of the data, critically reviewing the manuscript P.J.P.: conception and design of the study, interpretation of the data, drafting the manuscript

## Competing interests

The authors declare no competing interests.

## Ethics

All studies were done in accordance with the Declaration of Helsinki. Based on the Finnish biobank act, participants entered the FinnGen study by signing an informed consent for biobank research[12]. Alternatively, separate research cohorts, collected prior the Finnish Biobank Act came into effect (in September 2013) and start of FinnGen (August 2017), were collected based on study-specific consents and later transferred to the Finnish biobanks after approval by Fimea (Finnish Medicines Agency), the National Supervisory Authority for Welfare and Health. Recruitment protocols followed the biobank protocols approved by Fimea. The Coordinating Ethics Committee of the Hospital District of Helsinki and Uusimaa approved the FinnGen study protocol Nr HUS/990/2017. The FinnGen study is approved by Finnish Institute for Health and Welfare (permit numbers: THL/2031/6.02.00/2017, THL/1101/5.05.00/2017, THL/341/6.02.00/2018, THL/2222/6.02.00/2018, THL/283/6.02.00/2019, THL/1721/5.05.00/2019 and THL/1524/5.05.00/2020), Digital and population data service agency (permit numbers: VRK43431/2017-3, VRK/6909/2018-3, VRK/4415/2019-3), the Social Insurance Institution (permit numbers: KELA 58/522/2017, KELA 131/522/2018, KELA 70/522/2019, KELA 98/522/2019, KELA 134/522/2019, KELA 138/522/2019, KELA 2/522/2020, KELA 16/522/2020), Findata permit numbers THL/2364/14.02/2020, THL/4055/14.06.00/2020, THL/3433/14.06.00/2020, THL/4432/14.06.00/2020, THL/5189/14.06.00/2020, THL/5894/14.06.00/2020, THL/6619/14.06.00/2020, THL/209/14.06.00/2021, THL/688/14.06.00/2021, THL/1284/14.06.00/2021, THL/1965/14.06.00/2021, THL/5546/14.02.00/2020, THL/2658/14.06.00/2021, THL/4235/14.06.00/2021, Statistics Finland (permit numbers: TK-53-1041-17 and TK/143/07.03.00/2020 (earlier TK-53-90-20) TK/1735/07.03.00/2021, TK/3112/07.03.00/2021) and Finnish Registry for Kidney Diseases permission/extract from the meeting minutes on 4th July 2019. The Biobank Access Decisions for FinnGen samples and data utilized in FinnGen Data Freeze 12 include: THL Biobank BB2017_55, BB2017_111, BB2018_19, BB_2018_34, BB_2018_67, BB2018_71, BB2019_7, BB2019_8, BB2019_26, BB2020_1, BB2021_65, Finnish Red Cross Blood Service Biobank 7.12.2017, Helsinki Biobank HUS/359/2017, HUS/248/2020, HUS/430/2021 §28, §29, HUS/150/2022 §12, §13, §14, §15, §16, §17, §18, §23, §58, §59, HUS/128/2023 §18, Auria Biobank AB17-5154 and amendment #1 (August 17 2020) and amendments BB_2021-0140, BB_2021-0156 (August 26 2021, Feb 2 2022), BB_2021-0169, BB_2021-0179, BB_2021-0161, AB20-5926 and amendment #1 (April 23 2020) and it´s modifications (Sep 22 2021), BB_2022-0262, BB_2022-0256, Biobank Borealis of Northern Finland_2017_1013, 2021_5010, 2021_5010 Amendment, 2021_5018, 2021_5018 Amendment, 2021_5015, 2021_5015 Amendment, 2021_5015 Amendment_2, 2021_5023, 2021_5023 Amendment, 2021_5023 Amendment_2, 2021_5017, 2021_5017 Amendment, 2022_6001, 2022_6001 Amendment, 2022_6006 Amendment, 2022_6006 Amendment, 2022_6006 Amendment_2, BB22-0067, 2022_0262, 2022_0262 Amendment, Biobank of Eastern Finland 1186/2018 and amendment 22§/2020, 53§/2021, 13§/2022, 14§/2022, 15§/2022, 27§/2022, 28§/2022, 29§/2022, 33§/2022, 35§/2022, 36§/2022, 37§/2022, 39§/2022, 7§/2023, 32§/2023, 33§/2023, 34§/2023, 35§/2023, 36§/2023, 37§/2023, 38§/2023, 39§/2023, 40§/2023, 41§/2023, Finnish Clinical Biobank Tampere MH0004 and amendments (21.02.2020 & 06.10.2020), BB2021-0140 8§/2021, 9§/2021, §9/2022, §10/2022, §12/2022, 13§/2022, §20/2022, §21/2022, §22/2022, §23/2022, 28§/2022, 29§/2022, 30§/2022, 31§/2022, 32§/2022, 38§/2022, 40§/2022, 42§/2022, 1§/2023, Central Finland Biobank 1-2017, BB_2021-0161, BB_2021-0169, BB_2021-0179, BB_2021-0170, BB_2022-0256, BB_2022-0262, BB22-0067, Decision allowing to continue data processing until 31st Aug 2024 for projects: BB_2021-0179, BB22-0067, BB_2022-0262, BB_2021-0170, BB_2021-0164, BB_2021-0161, and BB_2021-0169, and Terveystalo Biobank STB 2018001 and amendment 25th Aug 2020, Finnish Hematological Registry and Clinical Biobank decision 18th June 2021, Arctic biobank P0844: ARC_2021_1001. The activities of the EstBB are regulated by the Human Genes Research Act. Individual level data analysis in EstBB was carried out under ethical approval 1.1-12/624 from the Estonian Committee on Bioethics and Human Research (Estonian Ministry of Social Affairs), using data according to release application 6-7/GI/33501 from the Estonian Biobank. An informed consent was obtained from participants in the NFBC1966/1986 cohort. The ethics committees of the Northern Ostrobothnia Hospital District in Oulu and the Helsinki University Hospital, Finland, approved the study plans.

## Additional information

## FinnGen

Aino Salminen [1] ✉, Kati Hyvärinen[2], Jarmo Ritari [2], Ulla Palotie [1], Markus Perola[6], Juha Sinisalo [11], Aki S. Havulinna [6,12,13], Päivi Mäntylä[7], Ulvi K. Gursoy[14], A. Liisa Suominen [7,15], David P. Rice [1], Vuokko Anttonen[16], Pekka Nieminen [1] & Pirkko J. Pussinen [1,7]

## Estonian Biobank Research Team

Kadri Reis[10], Anu Reigo [10] & Priit Palta [10]

A full list of members and their affiliations appears in the Supplementary Information.

