## [Transparent Peer Review file · Nature Communications]

Genome-wide association study of pulpal and apical diseases

Corresponding Author: Dr Aino Salminen

Version 0:

Reviewer comments:

Reviewer #1

(Remarks to the Author)

This paper describes a Genome wide Association study of apical periodontitis in a large cohort with further analyses looking at in silico prediction of variant/gene function. The study has merit particularly due to the large sample size however major concerns regarding other aspects of the study especially the distribution of the study groups based on the IDC codes without clinical phenotyping are significant. There are lots of analyses performed but there is no evidence that the identified genes are expressed in dental tissues of interest and that the variants are functional (or that would be in linkage disequilibrium with a functional variant). There are no functional characterization assays showing effect of such variants. The focus on the phenome-wide analysis also does not provide any meaningful interpretations.

Introduction:

1) While endodontic infections as a consequence of dental caries are among the causes of tooth loss, this readership would benefit from a more complete perspective of the main causes of tooth loss that also include periodontal disease.
2) The technical terms to describe patients' phenotypes are incorrect – for instance, there is no dental diagnostic term nor ICD code such as 'painful pulpitis'. The authors should use the correct diagnostic terminology adopted by the American Association of Endodontists and/or European Society of Endodontology and include appropriate definitions with lay terms when needed.

Methods:

1) cohorts description – some information is available however the authors do not report proportions of the ethnicities of study populations. While all cohorts were from Finland, ethnicities should be available for all participants. The proportions of the different ancestry groups within the cases and controls should be described in a Table.
2) ICD codes – the exact ICD code terms should be used. For example, the category endodontic infections stated to comprise all K04 codes – is actually named Diseases of Pulp and periapical tissues, and so on. Additional descriptions may follow to facilitate understanding but the proper terms should be used.
3) Incorrect phenotype categorization - Phenotype category of Pulpitis includes K04.08 (radicular cyst) and K04.09 (unspecified diseases of pulp and periapical tissues). These are not correct, specifically K04.08. Similarly, Phenotype category of necrosis or apical periodontitis includes K04.03 (abnormal hard tissue formation in pulp) which often times is associated with a vital pulp. These are critical flaws and may have skewed results.
4) Analyzing categories 2 and 3 sound more like a stratified analyses of category 1 than independent analysis.
5) Controls are individuals without K04 category diagnosis. What about periodontal disease as a confounder? These should have been excluded from the controls.
6) Clinical phenotyping – out of the 4 cohorts the Northern Finland one appears to have some evidence of actual clinical phenotyping. However, it is not clear why the authors defined proxy phenotypes of 1) deep caries AND pain and 2) Apical lesions OR pain were used. There are many cases of deep caries without associated pain, similarly there are many cases of apical lesions with pain (symptomatic apical periodontitis). Parogene cohort – apical rarefactions were registered from panoramic radiographs, which are not standard of care for diagnosis of endodontic lesions as they do not offer enough resolution. A Periapical radiograph or computed tomography would have been appropriate. What rarefaction sizes were considered for inclusion in the study and how were they truly evaluated to be of endodontic nature? There are individuals with pulpal necrosis/irreversible pulpitis with no periapical radiographic changes (something which is often seen clinical). What happens to these individuals? The lack of specific clinical phenotyping in the cohorts is a major concern.
7) Finngen cohort – how many records were from diagnoses using the hospital discharge registry (1995 and on) and how

many from the primary care?

8) "Regular" dental pain is mentioned – what does it mean and how was this scored? Was it at the time of diagnosis? Spontaneous or stimulated?

9) No description of matching of cases and controls. What steps were taken to ensure cohorts were independent of each other and no relatedness between cases and controls?

10) Please provide names and version of genotyping arrays

11) Please justify exclusion of variants with low minor allele count <3?

12) Please investigate if there are any systematic biases that may be present in the association results, e.g. by calculating the genomic inflation factor lambda? Not found in the paper.

Results:

1) Tables should include a column showing affected or nearby genes for each SNP listed and the frequency of alternate allele in cases and controls. Results for very low allele frequencies (0.02 and 0.04) deserve caution.

2) FDR = 0.049 or 0.048 should be rounded to 0.05 and the significance reevaluated

Various genome-wide significant associations were reported for each category - which was expected given that categories 2 and 3 are a subset of category 1. Were associated SNPs in LD with any potential functional variant in a relevant gene or a known gene?

3) The association with *HORMAD2*, a gene in which variants have been reported in individuals with Non-Suppurative Otitis Media and Male Infertility is definitely striking with P values up to 10⁻²². Whether *HORMAD2* is expressed in the disease relevant tissues however is unknown. The use of eQTLs is a good approach albeit confirmation of gene expression is needed to establish a gene in a given disease process.

4) Some experimental characterization procedures would benefit to establish relevance of the findings

5) Results for categories 2 and 3 reach astronomical p values (10⁻¹²⁰) that would require cautious interpretation given the questionable assignment of categories mentioned above

6) No sex-specific results described

Discussion

1) There is very little discussion of variant function with respect to the underlying biology of the endodontic categories studied or of any previously associated diseases

2) No mention of expression or lack thereof of expression of relevant genes in diseased tissues.

3) There is a lack of supportive references to some statements made (e.g. page 8 last sentence)

4) HLA genes are often hotspots – no discussion on this

Reviewer #2

(Remarks to the Author)

Genome-wide association study of endodontic infections

Interesting and important research.

Abstract:

The abstract accurately describes the short-summarized content of the article.

Key words are adequate in quality and number.

Introduction:

Information about dental caries, endodontic infections, and epidemiological data have been given in the introduction section.

Aim has been defined in this section.

Methods

Scientific type of the study, data resource, analysis method, statistical analysis methods have been defined in this section.

Results

The result section contains findings and observations in regards of the analysis. Figures and tables have been explained in detail in this section.

The authors also did a replication in others sub-cohorts.

Discussion

The discussion section states the importance and significance of the findings.

The authors have tried to explain the rationale of each association, and they tried to explain the genes function.

The authors should try to discuss some possible bias in this study, for example, recall bias.

Also, the authors should discuss something about the importance to understand the genetic factors and related with socio-environmental factors.

References:

The references section contains appropriate and number of references.

Figures:

Figures are demonstrative and clear.

Tables:

Tables are very well-designed.

Reviewer #3

(Remarks to the Author)

This is a well-written and comprehensive manuscript. The main strengths include the high statistical power, relatively unexplored phenotype(s) and replication of main findings.

My main comment relates to the visibility (or not) of genetic effects on caries in these analyses.

The results are presented as showing evidence of a SNP effect on susceptibility to endodontic infection, i.e SNP -> endodontic infection.

Endodontic infections usually result from caries (or less commonly, trauma or severe periodontal infection leading to a periodontal-endodontic lesion). Since caries is itself a heritable trait, I would expect caries risk variants to associate with endodontic infection, with the causal path involving SNP -> caries -> endodontic infection.

An alternative way of describing this - is that cases and controls are not matched on a heritable exposure (caries). Cases include those with caries and susceptible to endodontic infections.

Controls include those with low genetic liability to endodontic infections AND those who were not at risk as they did not have caries.

I do not think this prevents useful exploration and description of endodontic infections, but I would like to see clearer discussion of the other possible explanations for the observed SNP-> endodontic infection associations, including pathways involving caries.

In addition I have some minor comments/questions.

Introduction;

"Number of missing teeth is used as an epidemiological biomarker which reflects the experienced oral disease burden causing also irreversible and evident systemic damage".

Missing teeth can also reflect eg orthodontic extractions or hypodontia with no systemic health implications. Consider revising this statement.

"identified eight loci with a suggestive level of statistical significance".

In general - I do not think we should encourage discussion of 'suggestive' loci which can be simply explained by chance. The fact that a polygenic score trained on these loci was essentially uninformative also suggests these may have been chance findings.

Results;

"the population attributable risks (PARs) of the major risk alleles 90 varied from 0.095 to 0.145"

This does not appear plausible given that all variants genome wide reportedly explain < 2% of variation. Looking at the formula, it appears to be written for a quantitative phenotype with mean 0 and SD 1? While the results from SUSi are on a log odds ratio scale? Please check the scale. Also see comment on heritability estimation

I note some caries risk loci are represented in the results (HLA MTMR3 etc) - see comment on discussion. While the study reports that HLA haplotypes are not associated with caries (based on old papers) this is slightly misleading given the strong HLA association in caries GWAS.

In the genetic correlation analysis I missed any overlap with caries/periodontitis? These appear to be the most relevant traits to guide interpretation of what the SNP associations actually capture.

Discussion:

See main comment

The heritability estimate is described as 'moderate' but reported as <2%, which I would consider 'low'.

"This may be due to differences between 266 the phenotypes, since the earlier study examined participants with deep caries

and the cases suffered from apical periodontitis, whereas the controls of the present study were not selected based on caries status".

I think the comment that the participants were not selected on caries status is important for overall interpretation and should come a bit higher in the discussion.

"Clearly, the two disease stages, 'Pulpitis' and 'Necrosis of pulp or apical periodontitis', displayed different genetic profiles". This statement appears to contradict the statement earlier in the results that the studied traits had essentially the same genetic underpinnings - see this statement in results "Genetic correlations between the phenotypes 'Endodontic 182 infections', 'Pulpitis', and 'Necrosis of pulp or apical periodontitis' were high ($r_g=0.96-0.99$)"

"This, however, was considered to originate from the use of observed scale and not liability scale"

The authors suggest this is due to a discrepancy between the prevalence of endodontic infections on the observed and liability scale. I do not understand this comment since the population was not selected based on endodontic infections and the prevalence is presumably representative of the target adult population of Finland?

I did wonder whether the case:control imbalance in the analysis means the N supplied to LDSR was too large - the heritability estimates would appear a little bit higher if the effective N (accounting for imbalance) was used instead.

Methods:

PAR

See previous comment and check whether the scales (linear /log odds ratio) are correct

Tables

Table 2

See comments about scale

Version 1:

Reviewer comments:

Reviewer #1

(Remarks to the Author)

The authors have clarified several points raised in the first review. I commend the authors on extensive analysis and large samples sizes however major concerns with the clinical phenotyping remain that compromise validity of the findings.

One of the key elements in large scale genetic association studies is careful phenotyping to avoid spurious associations. As commented before, the use of panoramic radiographs is not standard of care for endodontic diagnosis of periapical lesions and therefore cannot be considered (Parogene cohort). Further, the Parogene cohort was focused on periodontal disease (periodontitis) and no endodontic diagnostic exams were available or proper periapical diagnosis therefore including this cohort in these analyses are inappropriate and misleading.

Much of the results interpretation are speculative suggesting functions for genes for which a role in oral disease is not relevant.

Lack of details hampering interpretation. For ex. discussion line 310 - 'only one of those showed significant association with our phenotype'...which one?

The stated findings that most of the associations between risk loci identified for endodontic infections were independent of caries raises additional concerns as endodontic infections are a continuum of dental caries (except in cases of dental trauma that may lead to pulp necrosis).

Outdated diagnostic terminologies - the term 'chronic apical periodontitis' was revised to 'asymptomatic apical periodontitis' in 2013.

As it relates to the potential associations between genetic studies of endodontic infections and systemic conditions, the authors fail to appreciate the body of knowledge supporting (or not) such associations.

Reviewer #3

(Remarks to the Author)

Thank you for the detailed response letter and comprehensive updates to the article.

I continue to think this is an interesting article with good statistical power, a relatively under-studied phenotype and independent replication of the main findings.

The authors have included major new sensitivity analyses exploring the possible mediating role of caries in these results, which addresses my previous question about this.

The authors have made various other updates following queries about the genetic correlation results and attributable risk calculations.

I am happy my previous queries have all been addressed and I have no new queries at this stage.

Version 2:

Reviewer comments:

Reviewer #1

(Remarks to the Author)

Thank you for addressing the remaining questions

REVIEWER COMMENTS

Reviewer #1 (Remarks to the Author):

This paper describes a Genome wide Association study of apical periodontitis in a large cohort with further analyses looking at in silico prediction of variant/gene function. The study has merit particularly due to the large sample size however major concerns regarding other aspects of the study especially the distribution of the study groups based on the IDC codes without clinical phenotyping are significant. There are lots of analyses performed but there is no evidence that the identified genes are expressed in dental tissues of interest and that the variants are functional (or that would be in linkage disequilibrium with a functional variant). There are no functional characterization assays showing effect of such variants. The focus on the phenome-wide analysis also does not provide any meaningful interpretations.

Introduction:

1) While endodontic infections as a consequence of dental caries are among the causes of tooth loss, this readership would benefit from a more complete perspective of the main causes of tooth loss that also include periodontal disease.

Response: We have revised the Introduction section according to the reviewer's suggestions (Introduction, second paragraph).

2) The technical terms to describe patients' phenotypes are incorrect – for instance, there is no dental diagnostic term nor ICD code such as 'painful pulpitis'. The authors should use the correct diagnostic terminology adopted by the American Association of Endodontists and/or European Society of Endodontology and include appropriate definitions with lay terms when needed.

Response: We have carefully checked the manuscript and made sure that we have used the international ICD10 coding and corresponding terminology. The term "painful pulpitis" in the Introduction was removed. We have changed the name of our first genotype to "Pulpal and apical diseases", which is a term used in the ESE guidelines.

Methods:

1) cohorts description – some information is available however the authors do not report proportions of the ethnicities of study populations. While all cohorts were from Finland, ethnicities should be available for all participants. The proportions of the different ancestry groups within the cases and controls should be described in a Table.

Response: In FinnGen, participants with non-Finnish genetic ancestry were removed from the analyses (Kurki et al, FinnGen provides genetic insights from a well-phenotyped isolated population. Nature. 2023;613(7944):508-518). This is mentioned in the Methods (page 16). In EstBB, 83% of the samples are from individuals of Estonian origin, 14% from individuals with Russian origin, and 3% from other ethnicities. In Parogene, all participants were of Finnish ethnicity. This information has been added in the Methods (page 15).

2) ICD codes – the exact ICD code terms should be used. For example, the category endodontic infections stated to comprise all K04 codes – is actually named Diseases of Pulp and periapical tissues, and so on. Additional descriptions may follow to facilitate understanding but the proper terms should be used.

Response: We have changed the name of our first genotype to "Pulpal and apical diseases", which is a term used in ESE guidelines. We have also revised the text as suggested (pages 4 and 5).

3) Incorrect phenotype categorization - Phenotype category of Pulpitis includes K04.08 (radicular cyst) and K04.09 (unspecified diseases of pulp and periapical tissues). These are not correct, specifically K04.08. Similarly, Phenotype category of necrosis or apical periodontitis includes K04.03 (abnormal hard tissue formation in pulp) which often times is associated with a vital pulp.

These are critical flaws and may have skewed results.

Response: We have double-checked the ICD10 diagnosis codes that were used to construct our phenotypes. The ICD codes that we used were based on the Finnish ICD10-library, and we confirmed that they were correct:

- Our phenotype “Pulpitis” did not include radicular cysts (K04.8) or unspecified diseases of pulp and periapical tissues (K04.9).
- Our phenotype “Necrosis or apical periodontitis” did not include abnormal hard tissue formation in pulp (K04.3).
- Our phenotype “Pulpitis” did include K04.08 Other pulpitis, which we think that should be included in this phenotype.
- Codes K04.01, K04.02, K04.03, K04.04, and K04.05 are not included in the Finnish ICD-10 code library. Thus, K04.03 was not used at all in the analyses (page 14, second paragraph).

4) Analyzing categories 2 and 3 sound more like a stratified analyses of category 1 than independent analysis.

Response: With phenotypes 2 and 3, we aimed at separating infections of vital and necrotic pulp. Indeed, there were differences between the phenotypes: The HLA locus and the loci with the lead SNP near *HORMAD2* and *RMI1* gene associated with all phenotypes (independently of caries). Loci near genes *PDE4B*, *IBTK*, *NR4A3*, and *VMP1* were associated with pulpitis, whereas loci near genes *LRP1B*, *TBCK*, and *MCPH1* were associated with necrosis of pulp or apical periodontitis.

5) Controls are individuals without K04 category diagnosis. What about periodontal disease as a confounder? These should have been excluded from the controls.

Response: We considered endodontic infections as a separate disease entity from periodontal diseases. The pathologic process in pulpitis and subsequent apical periodontitis and in periodontitis is very different, although both derive from a dysbiotic microbiome. In addition, the prevalence of combined endodontic-periodontal lesions is low even among patients with periodontitis: at patient level 4.9%, and at tooth level 0.4% (Ruetters M, Gehrig H, Kronsteiner D, Schuessler DL, Kim TS. Prevalence of endo-perio lesions according to the 2017 World Workshop on the Classification of Periodontal and Peri-Implant Disease in a university hospital. *Quintessence Int.* 2022 Jan 7;53(2):134-142.)

6) Clinical phenotyping – out of the 4 cohorts the Northern Finland one appears to have some evidence of actual clinical phenotyping. However, it is not clear why the authors defined proxy phenotypes of 1) deep caries AND pain and 2) Apical lesions OR pain were used. There are many cases of deep caries without associated pain, similarly there are many cases of apical lesions with pain (symptomatic apical periodontitis).

Response: In NFBC 1966 and 1986, there was no verified clinical diagnosis of pulpitis or any other pulpal pathology available. Thus, proxy phenotypes were created indicating participants with symptoms and/or findings related to pulpitis/pulpal pathology at the time of clinical examination or history of it: 1) clinical data on caries (ICDAS), which was used for defining deep caries lesions, 2) questionnaire data about oral/dental pain, and 3) panoramic x-rays with diagnoses of periapical lesions in NFBC 1966.

In the revised manuscript, we have modified the second phenotype for replication. Replications were done with phenotypes ‘Deep caries and regular dental pain’ based on ICDAS-classification and questionnaire (a proxy for pulpitis), and ‘Apical lesions’ based on panoramic x-ray (Table S2).

Parogene cohort – apical rarefactions were registered from panoramic radiographs, which are not standard of care for diagnosis of endodontic lesions as they do not offer enough resolution. A Periapical radiograph or computed tomography would have been appropriate. What rarefaction sizes were considered for inclusion in the study and how were they truly evaluated to be of endodontic nature? There are individuals with pulpal necrosis/irreversible pulpitis with no periapical radiographic changes (something which is often seen clinical). What happens to these individuals? The lack of specific clinical phenotyping in the cohorts is a major concern.

Response: The classification of apical lesions is based on our earlier publication (Liljestrand JM, Mäntylä P, Paju S, Buhlin K, Kopra KA, Persson GR, Hernandez M, Nieminen MS, Sinisalo J, Tjäderhane L, Pussinen PJ. Association of Endodontic Lesions with Coronary Artery Disease. *J Dent Res*. 2016;95(12):1358-1365). For this previous study, the number of teeth with widened periapical spaces and apical rarefactions were recorded by a radiologist specialized in oral and maxillofacial radiology. The definitions are in concordance with the commonly used periapical index (PAI); widened periapical spaces were scored as PAI = 3 and apical rarefaction as PAI = 4 to 5 (Ørstavik D, Kerekes K, Eriksen HM. 1986. The periapical index: a scoring system for radiographic assessment of apical periodontitis. *Endod Dent Traumatol*. 2(1):20–34).

Response: The reviewer is correct that the panoramic radiographs provide limited resolution to diagnose endodontic lesions. However, it was not possible to take periapical radiographs or computed tomographies of participants in the Parogene study because of limitations in the ethical permit. We have added this in the limitations section of the Discussion (page 12, second paragraph).

7) FinnGen cohort – how many records were from diagnoses using the hospital discharge registry (1995 and on) and how many from the primary care?

Response: We have added these numbers in the Methods (page 13, first paragraph). Of individuals with pulpal or apical diseases, 120,881 were identified from the primary care register, 1018 from hospital discharge registers, and 3 from the cause of death register.

8) “Regular” dental pain is mentioned – what does it mean and how was this scored? Was it at the time of diagnosis? Spontaneous or stimulated?

Response: The “regular dental pain” was not diagnosed in the clinical examination but was based on the questionnaire data, which was collected at the same time as the clinical examinations took place. The data on any pain in the mouth or teeth was collected with the same questions in both the NFBC 1966 and 1986 studies. Q1: “Was the reason for the previous dental visit pain?” with yes or no answers. Q2: “Have you felt any pain or discomfort in your mouth during the last month?” with six alternative answer options (very often, fairly often, sometimes, very rarely, never, I can’t say).

9) No description of matching of cases and controls. What steps were taken to ensure cohorts were independent of each other and no relatedness between cases and controls?

Response: In our GWAS in FinnGen, we used the REGENIE method, which accounts for sample relatedness and population structure (Mbatchou J et al. Computationally efficient whole-genome regression for quantitative and binary traits. *Nat Genet* 2021;53:1097–1103). NFBCs 1966 and 1986 and Parogene are included in FinnGen, and therefore individuals that belonged to these cohorts were removed before the discovery GWAS analyses.

10) Please provide names and version of genotyping arrays

Response: FinnGen samples were genotyped with ThermoFisher, Illumina, and Affymetrix arrays. Genome variant data from most of the samples was produced using a customised genotyping chip. The FinnGen ThermoFisher Axiom custom chip array contains 723,376 probe sets for 664,510 genetic markers. In addition to the core GWAS markers (about 500,000), the chip contains about 116,000 coding variants enriched in Finland, >10,000 specific markers for the HLA/KIR region, almost 15,000 ClinVar variants, about 4,600 pharmacogenomic variants and 57,000 selected markers that were of special interest for the partners. In addition, FinnGen includes existing genotypes from approximately 80 000 individuals from previous studies, which have used various generations of Illumina and Affymetrix GWAS arrays.

Information on genotyping is available here: <https://www.finnngen.fi/en/node/1996>. The link has been added on page 15.

11) Please justify exclusion of variants with low minor allele count < 3?

Response: This is part of the FinnGen quality control pipeline to account for sequencing errors or very low frequency variants.

12) Please investigate if there are any systematic biases that may be present in the association results, e.g. by calculating the genomic inflation factor lambda? Not found in the paper.

Response: Genomic inflation factors were $\lambda = 1.13$ for the phenotype “Pulpal and apical diseases”, $\lambda = 1.10$ for the phenotype “Pulpitis”, and $\lambda = 1.12$ for the phenotype “Necrosis or apical periodontitis”. These have been added in the Figure 1 legend. For polygenic traits, the LD Score regression intercept may be a more powerful and accurate correction factor than genomic control. The LD score regression intercepts were 1.0224 (SE 0.0097), 1.0414 (0.0101), and 1.0495 (0.01) for ‘Pulpal and apical diseases’, ‘Pulpitis’, and ‘Necrosis or apical periodontitis’, respectively.

Results:

1) Tables should include a column showing affected or nearby genes for each SNP listed and the frequency of alternate allele in cases and controls. Results for very low allele frequencies (0.02 and 0.04) deserve caution.

Response: The frequencies of alternative allele in cases and controls have been added (Table 2).

2) FDR =0.049 or 0.048 should be rounded to 0.05 and the significance reevaluated

Response: The threshold for statistical significance in the replication analyses was set to 0.05. Since $0.049 < 0.05$, we considered this statistically significant. Another criterion for successful replication was the effect direction, which was required to be the same as in the discovery analysis.

Various genome-wide significant associations were reported for each category - which was expected given that categories 2 and 3 are a subset of category 1.

Response: With phenotypes 2 and 3, we aimed at separating infections of vital and necrotic pulp. The supplementary figure presenting the frequencies of diagnosis codes (Figure S1) shows the differences between the phenotypes.

Were associated SNPs in LD with any potential functional variant in a relevant gene or a known gene?

Response: The LD structure and causal variants were analysed using the FinnGen finemapping pipeline, as described in the methods (page 17). Functionality was further analysed with in silico analyses (Table S8) and by compiling the Regulatory significance scores of most credible variants (Table S7). Among significant SNPs or SNPs in LD with them, we found two nonsynonymous missense and 32 UTR variants. These are included in Table S1 and reported in the text (page 6).

3) The association with *HORMAD2*, a gene in which variants have been reported in individuals with Non-Suppurative Otitis Media and Male Infertility is definitely striking with P values up to 10^{-22} . Whether *HORMAD2* is expressed in the disease relevant tissues however is unknown. The use of eQTLs is a good approach albeit confirmation of gene expression is needed to establish a gene in a given disease process.

Response: We analysed several whole genome RNAseq results from Gene Expression Omnibus database (<https://www.ncbi.nlm.nih.gov/geo/>) deriving from dental pulp tissue, which showed that *HORMAD2* gene was minimally expressed in adult pulp (see table below), and was not differentially expressed in pulpitis. In the same locus, expressions of several genes were affected by pulpitis (*KREMEN1*, *RFPL1S*, *MTMR3*, *LIF*, *OSM*, and *SF3A1*) and these are now listed in the manuscript in table S6.

Gene	Relative expression in dental pulp	Categorized expression	Up/downregulation in pulpitis
KREMEN1	0.61	med	0.58x
RFPL1S	0.016	min	1.6x
RFPL1	0.009	min	

NEFH	0.05	min	
THOC5	0.40	low	
NIPSNAP1	2.12	high	
NF2	1.04	med	
CABP7	0.13	low	
ZMAT5	1.12	med	
UQCR10	5.00	top	
ASCC2	0.91	med	
MTMR3	0.59	low	0.57x
MIR6818	0.0019	min	
HORMAD2-AS1	0.07	min	
HORMAD2	0.0001	min	
LIF-AS1	0.009	min	
LIF	1.75	high	3.3x
OSM	0.0029	min	5.9x
CASTOR1	0.6100	med	
TBC1D10A	0.94	med	
SF3A1	1.79	high	0.48x
SEC14L3	0	min	
DUSP18	0.13	low	

4) Some experimental characterization procedures would benefit to establish relevance of the findings

Response: We have added RNAseq results from the GEO database (see the response above) and reanalysed the associations between our lead SNPs and three phenotypes including adjustment for DMFS. This new data helped us to interpret the biological meaning of our results.

5) Results for categories 2 and 3 reach astronomical p values (10⁻¹²⁰) that would require cautious interpretation given the questionable assignment of categories mentioned above

Response: The lowest p-value in our GWAS results is 10⁻²², which we think is comparable to previous GWAS studies.

In Supplementary table S10, where we investigated the associations between our lead SNPs and other phenotypes, some of the p-values are indeed very low. These low p-values were found for the associations between SNPs located in the HLA region and autoimmune diseases, such as coeliac disease and type I diabetes. FinnGen includes ~500,000 participants and HLA is known to associate strongly with autoimmune diseases, and therefore, we do not find it surprising that the results are highly significant.

6) No sex-specific results described

Response: The sex-specific results are now added in a supplemental table (Table S4). Although the effect sizes and p-values differed in magnitude between males and females, most of the results had same effect direction and low p-values in both sexes.

Discussion

1) There is very little discussion of variant function with respect to the underlying biology of the endodontic categories studied or of any previously associated diseases

Response: We have added Discussion about the role of caries in the disease process (Discussion, paragraphs 1-3).

2) No mention of expression or lack thereof of expression of relevant genes in diseased tissues.

Response: We have added RNAseq results and reanalysed the associations including adjustment for DMFS (see response above).

3) There is a lack of supportive references to some statements made (e.g. page 8 last sentence)

Response: We have added references to support our statements in Discussion.

4) HLA genes are often hotspots – no discussion on this

Response: We have added discussion about this (page 10, first paragraph).

Reviewer #2 (Remarks to the Author):

Genome-wide association study of endodontic infections

Interesting and important research.

Abstract:

The abstract accurately describes the short-summarized content of the article.

Key words are adequate in quality and number.

Introduction:

Information about dental caries, endodontic infections, and epidemiological data have been given in the introduction section.

Aim has been defined in this section.

Methods

Scientific type of the study, data resource, analysis method, statistical analysis methods have been defined in this section.

Results

The result section contains findings and observations in regards of the analysis. Figures and tables have been explained in detail in this section.

The authors also did a replication in others sub-cohorts.

Discussion

The discussion section states the importance and significance of the findings.

The authors have tried to explain the rationale of each association, and they tried to explain the genes function. The authors should try to discuss some possible bias in this study, for example, recall bias.

Also, the authors should discuss something about the importance to understand the genetic factors and related with socio-environmental factors.

Response: We thank the reviewer for the supportive review of our manuscript. We have added these in the limitations part of the Discussion section (page 12, second paragraph).

References:

The references section contains appropriate and number of references.

Figures:

Figures are demonstrative and clear.

Tables:

Tables are very well-designed.

Reviewer #3 (Remarks to the Author):

This is a well-written and comprehensive manuscript. The main strengths include the high statistical power, relatively un-explored phenotype(s) and replication of main findings.

My main comment relates to the visibility (or not) of genetic effects on caries in these analyses.

The results are presented as showing evidence of a SNP effect on susceptibility to endodontic infection, i.e SNP -> endodontic infection.

Endodontic infections usually result from caries (or less commonly, trauma or severe periodontal infection leading to a periodontal-endodontic lesion). Since caries is itself a heritable trait, I would expect caries risk variants to associate with endodontic infection, with the causal path involving SNP -> caries -> endodontic infection.

An alternative way of describing this - is that cases and controls are not matched on a heritable exposure (caries). Cases include those with caries and susceptible to endodontic infections. Controls include those with low genetic liability to endodontic infections AND those who were not at risk as they did not have caries.

I do not think this prevents useful exploration and description of endodontic infections, but I would like to see clearer discussion of the other possible explanations for the observed SNP-> endodontic infection associations, including pathways involving caries.

Response: We thank the reviewer for pointing out this important aspect. We have reanalysed the associations including adjustment for DMFS, which was available for 84,690 individuals for the phenotype "Pulpal and apical diseases". These new data helped us to interpret the results and add understanding about the role of caries in the disease process. The results are presented in Table S4 and discussion is added on the role of caries in the disease process (page 10, first and second paragraph; page 11, first paragraph).

The HLA locus and the loci with the lead SNPs near HORMAD2 and RMI1 genes were associated with all phenotypes independently of caries. Loci near genes PDE4B, IBTK, NR4A3, and VMP1 were associated with pulpitis, whereas loci near genes LRP1B, TBCK, and MCPH1 were associated with necrosis and apical periodontitis independent of caries. Three loci lost the significance of the association (CAMD2, NOVA1, ZNF521) when the models were adjusted for caries.

In addition I have some minor comments/questions.

Introduction;

"Number of missing teeth is used as an epidemiological biomarker which reflects the experienced oral disease burden causing also irreversible and evident systemic damage".

Missing teeth can also reflect eg orthodontic extractions or hypodontia with no systemic health implications. Consider revising this statement.

Response: We have revised the section and removed the sentence about missing teeth as an epidemiological biomarker (Introduction, second paragraph).

"identified eight loci with a suggestive level of statistical significance".

In general - I do not think we should encourage discussion of 'suggestive' loci which can be simply explained by chance. The fact that a polygenic score trained on these loci was essentially uninformative also suggests these may have been chance findings.

Response: We have modified this sentence in the Introduction (page 3, third paragraph).

Results;

"the population attributable risks (PARs) of the major risk alleles 90 varied from 0.095 to 0.145"
This does not appear plausible given that all variants genome wide reportedly explain < 2% of variation. Looking at the formula, it appears to be written for a quantitative phenotype with mean 0 and SD 1? While the results from SUSi are on a log odds ratio scale? Please check the scale. Also see comment on heritability estimation

Response: We thank the reviewer for this comment. We had erroneously used OR:s instead of genotype specific risk ratios (RRs) in the calculation as described in the reference Witte et al., 2014. We have now estimated risk ratios with the assumption of Hardy-Weinberg-equilibrium among cases and controls and calculated the population attributable fractions according to Witte's formula varying from 0 to 1 and present them as risk percents (PARs). PARs are one way to estimate contribution on the population level. We agree that the results suggest large contributions which in case of chromosome 22 locus reflects the large allele frequency of the major risk allele.

I note some caries risk loci are represented in the results (HLA MTMR3 etc) - see comment on discussion. While the study reports that HLA haplotypes are not associated with caries (based on old papers) this is slightly misleading given the strong HLA association in caries GWAS.

Response: After adjusting the associations between our top SNPs and phenotypes with DMFS, the association between endodontic infections and HLA retained its significance. On the other hand, the same variants associating with endodontic infections showed an association with two caries phenotypes in FinnGen. These results are added in the revised manuscript (pages 5-6, tables S4 and S5) and we have added discussion about the role of HLA on all steps of disease progress (page 10, paragraph one).

In the genetic correlation analysis I missed any overlap with caries/periodontitis? These appear to be the most relevant traits to guide interpretation of what the SNP associations actually capture.

Response: We have added genetic correlations of pulpal and apical diseases and caries (Figure 2, page 11, table S15).

Discussion:

See main comment

"This may be due to differences between the phenotypes, since the earlier study examined participants with deep caries and the cases suffered from apical periodontitis, whereas the controls of the present study were not selected based on caries status".

I think the comment that the participants were not selected on caries status is important for overall interpretation and should come a bit higher in the discussion.

Response: We have revised the discussion about caries regarding the reviewer's aspect and the new results from the DMFS-adjusted analyses (pages 5-6).

"Clearly, the two disease stages, 'Pulpitis' and 'Necrosis of pulp or apical periodontitis', displayed different genetic profiles". This statement appears to contradict the statement earlier in the results that the studied traits had essentially the same genetic underpinnings - see this statement in results "Genetic correlations between the phenotypes 'Endodontic 182 infections', 'Pulpitis', and 'Necrosis of pulp or apical periodontitis' were high ($r_g=0.96-0.99$)"

Response: We have modified the sentence in the Discussion (page 11, third paragraph).

The heritability estimate is described as 'moderate' but reported as <2%, which I would consider 'low'.

"This, however, was considered to originate from the use of observed scale and not liability scale"

The authors suggest this is due to a discrepancy between the prevalence of endodontic infections on the observed and liability scale. I do not understand this comment since the population was not selected based on endodontic infections and the prevalence is presumably representative of the target adult population of Finland?

I did wonder whether the case:control imbalance in the analysis means the N supplied to LDSR was too large - the heritability estimates would appear a little bit higher if the effective N (accounting for imbalance) was used instead.

Response: We thank the reviewer for thoughtful comments regarding the heritability estimate and its interpretation. Below, we address your concerns point by point:

Moderate vs. Low Heritability: Reviewer noted that the heritability estimate of 2% was described as 'moderate,' considering it 'low.' We would like to clarify our interpretation. The heritability estimate for our phenotype was 2%, which is of the same magnitude as the heritability estimate for stroke (2%) in our analysis. Stroke is widely considered to be a moderately heritable trait, with estimates in the literature ranging between 38% and 39%. Therefore, we interpret our results as indicating that the heritability of our phenotype is also moderate, especially in the context of heritability estimates derived from the LDSC method.

Observed Scale vs. Liability Scale: The reviewer raised concerns about our explanation attributing the low heritability to the observed scale versus the liability scale. While it is true that the prevalence in our population sample reflects the general adult population and is representative, heritability estimates for binary traits are inherently sensitive to the scale on which they are analyzed. When analyzed on the observed scale, heritability is directly tied to the proportion of cases in the sample. In contrast, on the liability scale, heritability accounts for the underlying continuous distribution of liability in the population and adjusts for population prevalence. Thus, we consider that use of observed scale can lead to lower heritability estimates, even when the population prevalence is representative. The scale has now been mentioned in the table title as well as in the text (page 15, limitations).

Case-Control Imbalance and Effective Sample Size: The reviewer suggested that the case-control imbalance in the analysis may have led to an inflated sample size (N) being provided to LDSC, potentially underestimating heritability. We appreciate this important observation and have re-examined our methods in light of it. Indeed, LDSC heritability estimation is sensitive to the effective sample size, which accounts for the imbalance in case and control counts. If the effective sample size is not accurately represented, it can bias the heritability estimates downward.

Methods:

PAR

See previous comment and check whether the scales (linear /log odds ratio) are correct

Response: We have recalculated the results and corrected them in Table 2.

Tables

Table 2

See comments about scale

Response: The PAR values have been corrected in Table 2.

Reviewers' comments:

Reviewer #1 (Remarks to the Author):

The authors have clarified several points raised in the first review. I commend the authors on extensive analysis and large samples sizes however major concerns with the clinical phenotyping remain that compromise validity of the findings.

One of the key elements in large scale genetic association studies is careful phenotyping to avoid spurious associations. As commented before, the use of panoramic radiographs is not standard of care for endodontic diagnosis of periapical lesions and therefore cannot be considered (Parogene cohort). Further, the Parogene cohort was focused on periodontal disease (periodontitis) and no endodontic diagnostic exams were available or proper periapical diagnosis therefore including this cohort in these analyses are inappropriate and misleading.

RESPONSE: We have now removed the Parogene cohort from the manuscript. The Parogene cohort (n = 396) was used in our manuscript only for replication purposes. We have three other larger replication cohorts (EstBB n = 211320, NFBC n = 2312, FinnGen-replication n = 14705), and therefore, removing Parogene did not have a major effect on the validity of our results.

Much of the results interpretation are speculative suggesting functions for genes for which a role in oral disease is not relevant.

Lack of details hampering interpretation. For ex. discussion line 310 - 'only one of those showed significant association with our phenotype"...which one?

RESPONSE: We have added the SNP information in the Discussion: “We selected genetic variants from an earlier publication¹¹ for replication in our discovery GWAS, but only one (rs12800372 near *TPCN2*) of those showed a significant association with our Pulpitis phenotype.”

The stated findings that most of the associations between risk loci identified for endodontic infections were independent of caries raises additional concerns as endodontic infections are a continuum of dental caries (except in cases of dental trauma that may lead to pulp necrosis).

RESPONSE: Both diseases, endodontic infections or caries, have only a moderate genetic component. Thus, the genetics behind the diseases may be independent from each other or overlapping. For example, some genetic variations may predispose to the progression of caries lesions into endodontic infections, whereas other variants may be protective. Our GWAS shows how complicated the polygenic associations are and provides researchers new ideas for basic research exploring the pathogenic pathways.

Outdated diagnostic terminologies - the term 'chronic apical periodontitis' was revised to 'asymptomatic apical periodontitis' in 2013.

RESPONSE: The phenotypes in our discovery analysis were based on ICD10 codes used in Finland. The ICD10 diagnosis code K04.5 is called Chronic apical periodontitis in the Finnish ICD10 library. However, we have now checked the terminology used elsewhere in the manuscript and updated the term chronic apical periodontitis to asymptomatic apical periodontitis.

As it relates to the potential associations between genetic studies of endodontic infections and systemic conditions, the authors fail to appreciate the body of knowledge supporting (or not) such associations.

RESPONSE: We have added this in the Discussion (page 11, first paragraph).

Reviewer #3 (Remarks to the Author):

Thank you for the detailed response letter and comprehensive updates to the article.

I continue to think this is an interesting article with good statistical power, a relatively under-studied phenotype and independent replication of the main findings.

The authors have included major new sensitivity analyses exploring the possible mediating role of caries in these results, which addresses my previous question about this.

The authors have made various other updates following queries about the genetic correlation results and attributable risk calculations.

I am happy my previous queries have all been addressed and I have no new queries at this stage.

RESPONSE: We thank the reviewer for their supportive statement.